# Whole genome sequencing of metastatic colorectal cancer reveals prior treatment effects and specific metastasis features

Pauline A. J. Mendelaar [1,13], Marcel Smid [1,13], Job van Riet [1,2,3], Lindsay Angus [1], Mariette Labots[4,5], Neeltje Steeghs[5,6], Mathijs P. Hendriks [5,7], Geert A. Cirkel[5,8], Johan M. van Rooijen[5,9], Albert J. Ten Tije[5,10], Martijn P. Lolkema [1,5], Edwin Cuppen[11,12], Stefan Sleijfer[1,5], John W. M. Martens [1,5] & Saskia M. Wilting [1✉]

In contrast to primary colorectal cancer (CRC) little is known about the genomic landscape of metastasized CRC. Here we present whole genome sequencing data of metastases of 429 CRC patients participating in the pan-cancer CPCT-02 study (NCT01855477). Unsupervised clustering using mutational signature patterns highlights three major patient groups characterized by signatures known from primary CRC, signatures associated with received prior treatments, and metastasis-specific signatures. Compared to primary CRC, we identify additional putative (non-coding) driver genes and increased frequencies in driver gene mutations. In addition, we identify specific genes preferentially affected by microsatellite instability. CRC-specific 1kb-10Mb deletions, enriched for common fragile sites, and *LINC00672* mutations are associated with response to treatment in general, whereas *FBXW7* mutations predict poor response specifically to EGFR-targeted treatment. In conclusion, the genomic landscape of mCRC shows defined changes compared to primary CRC, is affected by prior treatments and contains features with potential clinical relevance.

[1] Department of Medical Oncology, Erasmus MC Cancer Institute, Erasmus University Medical Center Rotterdam, Rotterdam, The Netherlands. [2] Cancer Computational Biology Center, Erasmus MC Cancer Institute, Erasmus University Medical Center Rotterdam, Rotterdam, The Netherlands. [3] Department of Urology, Erasmus MC Cancer Institute, Erasmus University Medical Center Rotterdam, Rotterdam, The Netherlands. [4] Department of Medical Oncology, Cancer Center Amsterdam, Amsterdam UMC, Vrije Universiteit Amsterdam, Amsterdam, The Netherlands. [5] Center for Personalized Cancer Treatment, Rotterdam, The Netherlands. [6] Department of Medical Oncology, The Netherlands Cancer Institute, Antoni van Leeuwenhoek, Amsterdam, The Netherlands. [7] Department of Medical Oncology, Northwest Clinics, Alkmaar, The Netherlands. [8] Department of Medical Oncology, Meander Medical Center, Amersfoort, The Netherlands. [9] Department of Medical Oncology, Martini Hospital, Groningen, The Netherlands. [10] Department of Medical Oncology, Amphia Hospital, Breda, The Netherlands. [11] Center for Molecular Medicine and Oncode Institute, University Medical Center Utrecht, Utrecht, The Netherlands. [12] Hartwig Medical Foundation, Amsterdam, The Netherlands. [13] These authors contributed equally: Pauline A.J. Mendelaar, Marcel Smid. ✉email: s.wilting@erasmusmc.nl

Primary colorectal cancer (CRC) can be divided into a major group of chromosomally instable tumors and a minor group of hypermutated, chromosomally stable tumors due to microsatellite instability (MSI) or *POLE* mutations[1]. Parallel to the described genomic subtype division, transcriptomic analysis was used to identify four consensus molecular subtypes (CMSs) with distinguishing features including prognosis[2].

Molecular analysis of CRC revealed specific genetic alterations with clinical implications. Mutations in *KRAS* and *BRAF* predict failure to treatment with EGFR-inhibitors, whereas copy number alterations of *ERBB2* or *IGF2*, and the occurrence of chromosomal translocations leading to fusion genes such as *NAV2/TCF7L1*, are potentially drug targetable[1,3].

Although the molecular knowledge of primary CRC has contributed to a better understanding of its pathogenesis, cancer-related mortality usually occurs as a consequence of distant metastases, in which ongoing mutational processes and selective treatment pressure can result in altered molecular characteristics[4].

To date, in-depth analyses of large series of colorectal cancer metastases are limited to studies using either whole-exome sequencing (WES) or targeted sequencing of cancer-associated genes[4–6]. Although these studies yielded extensive knowledge on the presence of specific genomic aberrations in mCRC, they do not necessarily reflect its complete molecular landscape. For optimal identification of mutational signatures, the power provided by whole-genome sequencing (WGS) data greatly exceeds that of WES[7]. Next to this, WGS simultaneously allows for the determination of MSI, structural rearrangements, chromothripsis, and kataegis. In addition, clinically relevant genetic alterations within noncoding regions were recently reported in primary CRC[8]. To date, the only other study which reported in detail on WGS data of colorectal metastases included 12 patients[4].

Here, we provide a comprehensive description of the molecular landscape of metastatic CRC (mCRC). We use WGS data obtained from a large multicenter, prospective collection of snap-frozen metastatic tissue biopsies from 429 patients starting a new line of systemic treatment[9]. In addition, matched RNA-seq data are available for 91 patients. The observed metastatic molecular landscape is compared to WGS data of primary CRC cohorts (Supplementary Table 1), associated with prior treatments as well as treatment response, and evaluated for clinical utility.

## Results

### Cohort description.
Clinical characteristics of our included cohort of 429 patients are summarized in Table 1. Median tumor purity (0.53 (IQR 0.38–0.67) was estimated on the obtained sequencing data and was not significantly different between biopsy sites. Based on a previously described WGS data analysis algorithm[9] 14 samples (3%) were scored as microsatellite instable (MSI), which is in concordance with the observed MSI frequency in mCRC in literature (4%)[10].

Based on the treatment data, the cohort can be divided in patients who did ($n = 284$) and who did not ($n = 124$) receive any systemic treatment prior to the moment the biopsy was taken. Within the group of prior-treated patients, 13 different combinations of treatment regimens were defined as specified in the materials and methods and listed in Table 1.

For 91 cases RNA-seq data were available, allowing us to determine their Consensus Molecular Subtype (CMS). Remarkably, using the CMS-classifier package, none of the metastatic CRC samples were classified as CMS3, whereas 10 were classified as CMS1, 41 as CMS2, and 14 as CMS4. The remaining 26 samples (29%) could not be classified into one of the 4 subtypes, which might be partly due to the presence of normal cells of noncolon origin in our metastatic setting. Indeed, using

the alternative CMSCaller algorithm, which is less dependent on signals from the tumor microenvironment, reduced the number of unclassified samples to 14 (15%), whereas still only 3 samples were classified as CMS3[11]. Twenty-two samples were classified as CMS1, 25 as CMS2, 3 as CMS3, and 27 as CMS4.

Regardless of the calling algorithm used, the estimated tumor cell percentage was significantly lower in biopsies classified as CMS4 than in the other subtypes (medians CMS1: 52.5 and 45%; CMS2 61 and 61%; CMS3: none and 66% and CMS4: 34.5 and 42%; KWH; $p = 0.0007$ and $p = 0.0156$ for CMS Classifier and CMSCaller, respectively), which is concordant with the described high-stroma content in this subtype[2].

### The molecular landscape of mCRC.
From the WGS data of all 429 cases, we distilled somatically acquired single nucleotide variants (SNVs), multiple nucleotide variants (MNVs), structural variants (SVs), insertions/deletions (InDels), and copy number variants (CNVs). The overall tumor mutational burden (TMB) representing the amount of SNVs, MNVs and InDels per Megabase (Mb), ranged from 0.96 to 366.15 with a median of 7.01 (95% CI 6.62–7.47). Using GISTIC2.0, we identified 55 recurrent CNVs (29 gains and 26 losses) within our entire cohort, containing a number of already known and putative driver genes (Supplementary Data 1). Chromothripsis was observed in 47 cases (11%), whereas kataegis was observed in 102 cases (24%), involving just a single chromosomal region in two-third of cases, with a maximum of 10 regions in one single case. Presence of kataegis was associated with MSI and high TMB (≥10; test for trend $p = 0.00014$). In fact, 9 out of 13 MSI cases had at least two kataegis regions.

We further evaluated the type and size of SVs observed in our cohort (Fig. 1). A broad range of differently sized Tandem Duplications (TD; ~14–93 kb) with a peak at 26 kb was observed, which was clearly distinct from the TD sizes previously observed in other cancers (~11 kb in *BRCA1*-mutated, ~231 kb in *CCNE1*-activated, and ~1.7 Mb TDs in *CDK12*-mutated cancer, respectively)[12]. Inversions in mCRC are usually over 10 Mb in size, while deletions range from ~10 kb to 1 Mb, with a distinct peak at ~128 kb. Events within this latter peak include many recurrent deletions in known Common Fragile Site (CFS) genes: e.g., *FHIT*, *RBFOX1*, and *MACROD2*. This phenomenon involving frequent deletions of CSF genes was recently described in primary CRC as well[13].

Using the ratio of nonsynonymous to synonymous substitutions caused by the somatic nucleotide mutations (SNV and InDels; dN/dS analysis), 23 genes were identified as putative driver genes (q < 0.05, Fig. 2, Table 2). In 99.1% of cases (425 out of 429) at least one of these 23 putative driver genes was mutated. Testing for mutual exclusivity only revealed already known associations: *KRAS* with *BRAF/NRAS/RNF43/TP53* (q = 1.06E-7, q = 1.54E-4, q = 0.004, and q = 0.017, respectively), and *APC* with *RNF43/BRAF* (both q = 1.54E-4; Supplementary Fig. 1). For those genes also present in the targeted panel used by Yaeger et al.[6], comparable mutation frequencies were observed in both cohorts (Table 2).

Similarly, for 15 noncoding genes an enriched mutation rate was observed compared to surrounding nonannotated regions (Table 3), suggesting these genes are relevant for the oncogenic process. These noncoding genes include *PTENP1*, a known tumor suppressor in CRC[14], *MALAT1*, for which an increased mutation rate was already described in a pan-cancer analysis[15], and *LINC00672*, described to promote chemo-sensitivity[16].

To further investigate the mechanisms underlying the observed SNVs and MNVs, we used the latest COSMIC mutational signatures (v3) to establish the presence and contribution of these

**Table 1 Cohort description.**

| Patient details | | Number of patients |
|---|---|---|
| Total cohort | | 429 |
| Gender | Female | 174 |
| | Male | 255 |
| Age (median (IQR,Range)) | | 64 (IQR 56–72, range 25–88) |
| Prior-treatment details | | |
| Systemic prior treatment: yes | | 284 |
| Treatment regimen | Manuscript code | |
| 5-FU/capecitabine–oxaliplatin doublet (CAPOX, FOLFOX) | PLAT/PYR | 121 |
| +bevacizumab | PLAT/PYR + targeted | 134 |
| 5-FU—Topisomerase inhibitor doublet (Irinotecan based, FOLFIRI) | TOP/PYR | 26 |
| +bevacizumab/panitumumab | TOP/PYR + targeted | 9 |
| 5-FU/capecitabine monotherapy | PYRmon | 39 |
| +bevacizumab | PYR + targeted | 36 |
| Topoisomerase inhibitor (Irinotecan) monotherapy | TOPmono | 67 |
| +bevacizumab | TOP + targeted | 7 |
| Oxaliplatin + bevacizumab/panitumumab | PLAT + targeted | 5 |
| Panitumumab/cetuximab/encorafenib+binimetinib/bevacizumab/regorafenib | Targeted mono | 35 |
| 5-FU/capecitabine–oxaliplatin–irinotecan triplet (FOLFOXIRI) | CHEMCOM | 2 |
| +bevacizumab | CHEMCOM + targeted | 5 |
| Other | Diverse | 15 |
| Systemic prior treatment: no | | 124 |
| Systemic prior treatment: unknown | | 21 |
| Radiotherapeutic prior treatment: yes | | 109 |
| RT + systemic treatment | | 68 |
| Chemoradiation | | 33 |
| Radiotherapy only | | 8 |
| Radiotherapeutic prior treatment: no | | 299 |
| Radiotherapeutic prior treatment: unknown | | 21 |
| Biopsy details | | Number of patients |
| Origin | | |
| | Liver | 287 |
| | Soft tissue | 84 |
| | Lymph node | 24 |
| | Lung | 21 |
| | Other | 13 |
| Technical details | | |
| | Tumor purity | 0.53 (IQR 0.38–0.67) |
| | Read coverage (median) | 102.6× (IQR 94.6×–112.0×) |

Patient Characteristics. PLAT/PYR (5-FU/capecitabine–oxaliplatin doublet (CAPOX, FOLFOX)), PLAT/PYR + targeted (bevacizumab added), TOP/PYR + targeted (bevacizumab added), PYRmono (5-FU/capecitabine monotherapy), PYR + targeted (bevacizumab added), TOP/PYR (5-FU-Topisomerase inhibitor doublet (Irinotecan based)), TOPmono (Topoisomerase inhibitor (Irinotecan) monotherapy), TOP + targeted (bevacizumab added), PLAT + targeted (Oxaliplatin + bevacizumab/panitumumab), Targeted mono (panitumumab, cetuximab, encorafenib + binimetinib, bevacizumab, regorafenib), CHEMCOM (5-FU/capecitabine–oxaliplatin–irinotecan triplet (FOLFOXIRI)), CHEMCOM + targeted (bevacizumab added), Other (diverse).

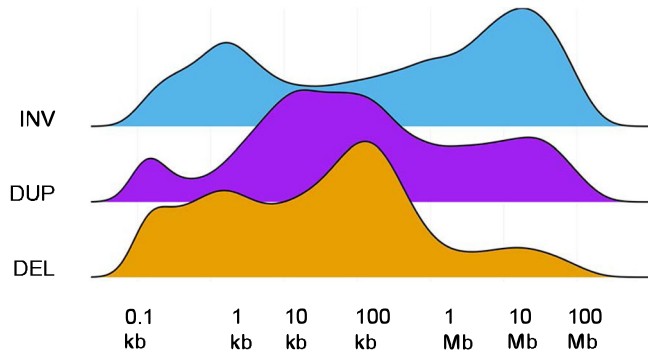

**Fig. 1 Size distributions of the different types of structural variants.** Ridge-plot of the density of genomic sizes of structural variants in metastatic CRC. INV inversions (blue), DUP tandem duplications (purple), DEL deletions (orange). Source data are provided as a Source Data file.

predefined mutational signatures in metastatic CRC[17]. We identified 11 single base signatures (SBS) and 9 double base signatures (DBS) that had a relative contribution of at least 10% in minimally 10 cases and as such were considered dominant signatures in mCRC; SBS1, SBS5, SBS8, SBS9, SBS17b, SBS18, SBS35, SBS39, SBS40, SBS41, SBS44, DBS2-9, and DBS11. De novo signature calling using the Non-negative Matrix Factorization algorithm (NMF)[18] did not identify additional signatures besides the known COSMIC signatures in our cohort.

**Effects of systemic prior treatment on the genomic landscape.** Patients receiving prior systemic treatment ($n = 284$) showed a significantly higher TMB, a higher number of SVs, a higher number of affected GISTIC CNV regions (7.58 versus 5.82; 208 versus 148; 31 versus 28, respectively; MWU $p$-values < 0.005), and more frequent occurrence of chromothripsis (6.5 versus 13.4%; Fisher exact test $p = 0.042$) compared to patients ($n = 124$) without prior systemic treatment. More specifically, we observed altered relative contributions for several mutational signatures in defined prior-treatment groups compared to treatment-naive patients ($n = 124$, Fig. 3 and Supplementary Data 2; MWU, FDR $p <$ =5.15E-7). Patients who were prior-treated with a combination therapy of PLAT/PYR + target showed increased relative contributions of SBS8, SBS17b, SBS35, and DBS5 compared to treatment-naive patients. These results

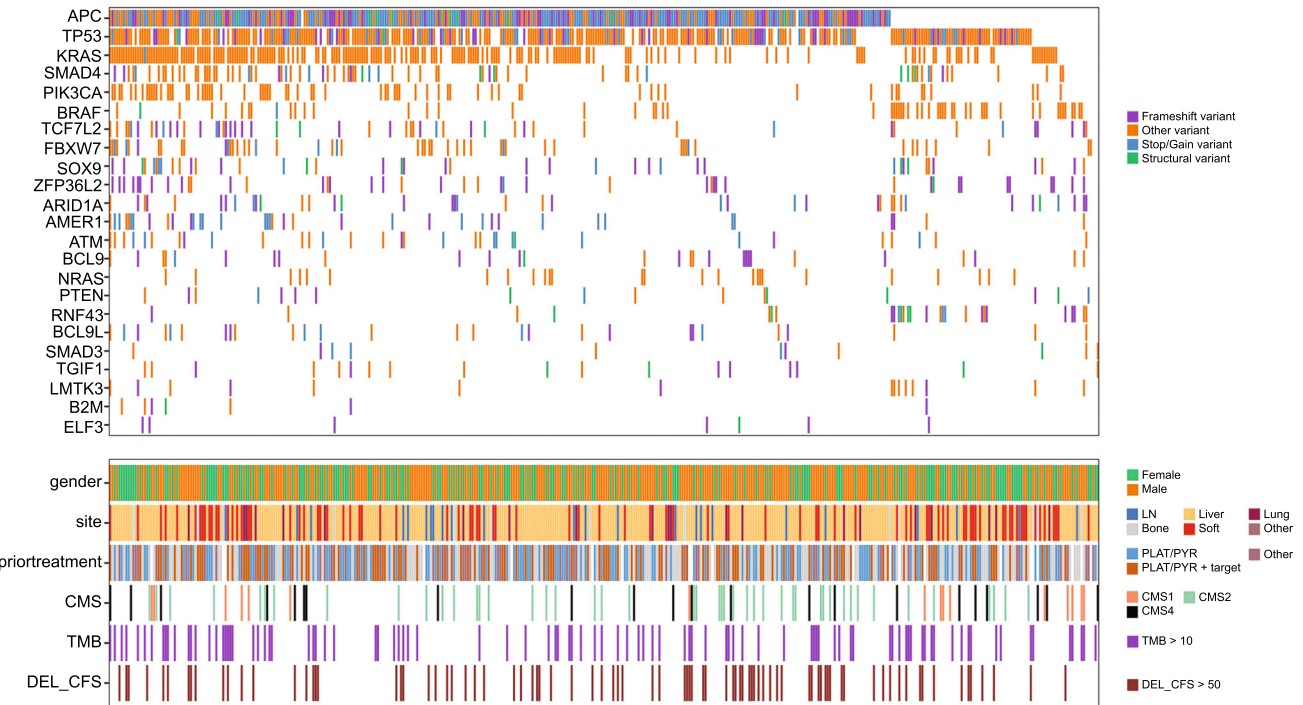

**Fig. 2 Oncoplot of metastatic CRC depicting identified driver genes and somatic mutations (SNV, InDels, and MNV).** Top panel: genes identified by dN/dS as driver genes per type of mutation; purple: frameshift variant; orange: other variant; blue: stop/gain variant; green: structural variant. Bottom panel: first track: clinical information: sex (male: orange; female: green) and second track: biopsy site. Track three (PLAT/PYR ± targeted) indicates which patients have been treated with platinum-based therapy (PLAT; e.g., oxaliplatin) and a pyrimidine-targeting drug (PYR; e.g., 5-FU), with or without the addition of another targeted treatment (±targeted; e.g., bevacizumab). Tracks four to six depict the distribution of the consensus molecular subtypes (CMS), tumor mutational burden (TMB), and the number of structural variant deletions of size 10kb–1Mb (DEL_CFS), partly associated with Common Fragile Sites (CFS), respectively. Source data are provided as a Source Data file.

are supported by previous studies in which DBS5 and SBS35 signatures were linked to the effect of platinum (PLAT) compounds, while SBS17b was detected specifically in 5-FU or capecitabine (PYR) exposed tumors[19]. SBS8 was previously indirectly associated with prior platinum treatment in metastatic breast cancer[17,20].

Remarkably, even though TMB was increased in patients who received prior treatment compared to treatment-naive patients, no specific mutations (coding or noncoding) were associated with any of the defined prior-treatment groups or with prior treatment in general. With regard to the GISTIC-defined CNVs, we found increased frequencies of gains at 6p22.1, 6p21.1, and 18p11.32 as well as losses at 3p14.2 and 8p21.3 in patients who received prior treatment (Supplementary Table 2; chi-square FDR < 0.05). More specifically, gains of 6p22.1 and 6p21.1 were also associated with a prior-treatment regimen containing PLAT/PYR ± target whereas loss at 8p21.3 was only associated with PLAT/PYR + target.

**Comparing metastatic CRC to primary CRC.** The above described characteristics of our metastatic cohort were related to previous reports on primary CRC to identify changes potentially linked to the metastatic process (Supplementary Table 1). Therefore, we compared the observed relative contributions of the 20 dominant mutational signatures in our cohort to primary CRC data described by Alexandrov et al. (PCAWG cohort)[17]. For this analysis only the 124 untreated metastatic CRC cases from our cohort were included, since multiple treatments are known to specifically affect these mutational signatures[17,19,20]. SBS1, 8 and 41, as well as DBS2, 4, and 6 showed a significantly increased relative contribution in untreated metastatic cases (MWU, FDR ≤ 0.01; Fig. 4), suggesting they may be associated with the

metastatic process. Etiologies for these signatures are either unknown (SBS8/41, DBS1) or appear age-related (SBS1, DBS2/DBS4), although DBS2 has also been linked to exposure to tobacco smoking and other endogenous and exogenous mutagens. Mutation frequencies per gene were compared between primary CRC (TCGA-DFCI cohort) and our total metastatic cohort. For this purpose, we selected genes mutated in primary CRC (TCGA-DFCI cohort) with >5% prevalence and complemented these with here identified metastatic driver genes regardless of their prevalence in primary CRC. Increased frequencies were only observed in driver genes *TP53*, *ZFP36L2*, *KRAS*, and *APC* (Fisher exact test, FDR ≤ 0.012). A decreased frequency was observed for 21 non-driver genes (Supplementary Table 3) and 1 driver gene, namely *PIK3CA* (Table 2). With respect to the identified putative noncoding drivers (Table 3), all of them were enriched in mCRC compared to primary CRC, except for *PIPSL* and *PTENP1* (ICGC dataset; Fisher exact test, FDR < 5.74E-4).

**Distinct mutational signature patterns in mCRC patients.** Unsupervised hierarchical clustering using the 20 dominant mutational signatures complemented with mutational signatures previously described in primary CRC (SBS15/17a/28/37 and DBS10), and mutational signatures showing a dominant relative contribution (>25%) in at least one of our samples (SBS10a/10b), revealed three major and three minor groups of patients (Fig. 5).

The three major groups are found in cluster 1, cluster 3, and cluster 6. Clusters 1 and 6 are labeled "prior treatment" and "primary-like" as they are enriched for either patients with or without prior treatment compared to all other clusters (Fisher's exact test: p = 4.588E-25 and p = 4.754E-15, respectively) and are

**Table 2 Mutation frequency driver genes.**

| Gene | dN/dS q-value | Metastatic CRC | | Primary CRC—(TCGA, combined studies cBioportal) | | | | % change in meta | Metastatic CRC—(Yaeger et al.) | | | |
|---|---|---|---|---|---|---|---|---|---|---|---|---|
| | | Mutations (N) | Mutations (%) | Mutations (N) | Mutations (%) | Fisher p-value | FDR Hochberg | | Mutations (N) | Mutations (%) | Fisher p-value | FDR Hochberg |
| TP53 | 0 | 317 | 73.9 | 1123 | 57.6 | 2.04E-10 | 4.7E-09 | 16.3 | 246 | 76.6 | 0.395 | 1 |
| ZFP36L2 | 0 | 42 | 9.8 | 97 | 5.0 | 3.61E-04 | 0.008 | 4.8 | Not present | | | |
| KRAS | 0 | 203 | 47.3 | 744 | 38.2 | 5.88E-04 | 0.012 | 9.1 | 127 | 39.6 | 0.037 | 0.487 |
| APC | 0 | 336 | 78.3 | 1372 | 70.4 | 8.86E-04 | 0.018 | 7.9 | 241 | 75.1 | 0.335 | 1 |
| PIK3CA | 0 | 68 | 15.9 | 445 | 22.8 | 0.001 | 0.023 | −7.0 | 49 | 15.3 | 0.840 | 1 |
| B2M | 5.37E-03 | 8 | 1.9 | 91 | 4.7 | 0.007 | 0.128 | −2.8 | 2 | 0.6 | 0.202 | 1 |
| SMAD4 | 0 | 74 | 17.2 | 243 | 12.5 | 0.010 | 0.164 | 4.8 | 47 | 14.6 | 0.367 | 1 |
| ATM | 9.50E-04 | 33 | 7.7 | 227 | 11.6 | 0.017 | 0.266 | −4.0 | 18 | 5.6 | 0.306 | 1 |
| FBXW7 | 0 | 51 | 11.9 | 301 | 15.4 | 0.061 | 0.912 | −3.6 | 25 | 7.8 | 0.068 | 0.814 |
| AMER1 | 0 | 37 | 8.6 | 209 | 10.7 | 0.220 | 0.912 | −2.1 | 11 | 3.4 | 0.004 | 0.067 |
| ARID1A | 1.13E-09 | 39 | 9.1 | 201 | 10.3 | 0.480 | 0.913 | −1.2 | 15 | 4.7 | 0.022 | 0.333 |
| BCL9 | 5.27E-02 | 28 | 6.5 | 107 | 5.5 | 0.419 | 0.913 | 1.0 | Not present | | | |
| BCL9L | 2.24E-06 | 27 | 6.3 | 133 | 6.8 | 0.750 | 0.913 | −0.5 | Not present | | | |
| BRAF | 0 | 56 | 13.1 | 273 | 14.0 | 0.644 | 0.913 | −1.0 | 38 | 11.8 | 0.657 | 1 |
| ELF3 | 5.37E-03 | 15 | 3.5 | 51 | 2.6 | 0.299 | 0.913 | −1.0 | Not present | | | |
| LMTK3 | 1.33E-02 | 7 | 1.6 | 56 | 2.9 | 0.530 | 0.913 | 0.6 | not present | | | |
| NRAS | 0 | 26 | 6.1 | 125 | 6.4 | 0.913 | 0.913 | −0.4 | 14 | 4.4 | 0.329 | 1 |
| PTEN | 4.30E-08 | 17 | 4.0 | 123 | 6.3 | 0.069 | 0.913 | −2.3 | 14 | 4.4 | 0.854 | 1 |
| RNF43 | 2.20E-02 | 28 | 6.5 | 162 | 8.3 | 0.239 | 0.913 | −1.8 | 21 | 6.5 | 1.000 | 1 |
| SMAD3 | 5.68E-03 | 11 | 2.6 | 72 | 3.7 | 0.309 | 0.913 | −1.1 | 11 | 3.4 | 0.518 | 1 |
| SOX9 | 0 | 41 | 9.6 | 177 | 9.1 | 0.782 | 0.913 | 0.5 | 16 | 5.0 | 0.025 | 0.352 |
| TCF7L2 | 1.07E-09 | 50 | 11.7 | 177 | 9.1 | 0.103 | 0.913 | 2.6 | 19 | 5.9 | 0.007 | 0.116 |
| TGIF1 | 1.33E-02 | 18 | 4.2 | 62 | 3.2 | 0.300 | 0.913 | 1.0 | Not present | | | |

Twenty-three genes identified as putative driver genes using the ratio of nonsynonymous to synonymous substitutions caused by the somatic nucleotide mutations (SNV and InDels; dN/dS analysis). P-values are derived from the Fisher exact test (two-sided) and corrected for multiple testing using the FDR (Hochberg) method.

**Table 3 Mutation frequency noncoding genes.**

| ENSG | Symbol | Size | Chr | Type | Metastatic CRC | | Mutation rate | FDR Hochberg | Primary CRC - (ICGC) | | Fisher p-value | FDR Hochberg |
|---|---|---|---|---|---|---|---|---|---|---|---|---|
| | | | | | Mutations (N) | Mutations (%) | | | Mutations (N) | Mutations (%) | | |
| ENSG00000273001 | AL731533.2 | 577 | 10 | lncRNA | 6 | 1.4 | 0.067591 | 0 | 0 | 0 | 0.001291 | 3.87E-03 |
| ENSG00000280325 | AC074183.2 | 921 | 7 | TEC | 25 | 5.8 | 0.033659 | 0 | a | a | | |
| ENSG00000261584 | AL513548.1 | 1723 | 6 | lncRNA | 14 | 3.3 | 0.012768 | 9.93E-24 | 3 | 0.3 | 3.84E-05 | 2.09E-04 |
| ENSG00000259834 | AL365361.1 | 3480 | 1 | lncRNA | 17 | 4.0 | 0.007184 | 2.48E-09 | 0 | 0 | 5.62E-09 | 7.30E-08 |
| ENSG00000264920 | AC018521.5 | 4583 | 17 | lncRNA | 16 | 3.7 | 0.00851 | 1.02E-08 | 1 | 0.1 | 2.06E-07 | 1.85E-06 |
| ENSG00000231784 | DBIL5P | 2676 | 17 | Transcribed_unitary_pseudogene | 18 | 4.2 | 0.008595 | 6.26E-07 | 0 | 0.0 | 2.39E-08 | 2.63E-07 |
| ENSG00000273033 | LINC02035 | 5475 | 3 | lncRNA | 23 | 5.4 | 0.006758 | 9.54E-06 | 1 | 0.0 | 6.15E-12 | 8.61E-11 |
| ENSG00000266979 | LINC01180 | 3985 | 17 | lncRNA | 13 | 3.0 | 0.005521 | 1.18E-05 | 1 | 0.1 | 5.00E-06 | 4.00E-05 |
| ENSG00000272070 | AC005618.1 | 3147 | 5 | lncRNA | 18 | 4.2 | 0.006991 | 4.19E-05 | 1 | 0.1 | 2.39E-08 | 2.63E-07 |
| ENSG00000251562 | MALAT1 | 8828 | 11 | lncRNA | 30 | 7.0 | 0.005551 | 0.000283 | 19 | 2.2 | 4.19E-05 | 2.09E-04 |
| ENSG00000261094 | AC007066.2 | 2710 | 9 | lncRNA | 11 | 2.6 | 0.008118 | 0.000287 | 2 | 0.2 | 1.86E-04 | 7.43E-04 |
| ENSG00000263874 | LINC00672 | 4216 | 17 | Protein_coding | 14 | 3.3 | 0.005455 | 0.000493 | 2 | 0.2 | 9.70E-06 | 6.79E-05 |
| ENSG00000180764 | PIPSL | 3349 | 10 | Transcribed_processed_pseudogene | 14 | 3.3 | 0.006569 | 0.0013 | 34 | 3.9 | 0.640 | 0.640 |
| ENSG00000237984 | PTENP1 | 3995 | 9 | Transcribed_processed_pseudogene | 15 | 3.5 | 0.005507 | 0.0013 | 22 | 2.5 | 0.376 | 0.640 |
| ENSG00000240859 | AC093627.4 | 5916 | 7 | lncRNA | 18 | 4.2 | 0.006085 | 0.0048 | 2 | 0.2 | 1.66E-07 | 1.66E-06 |

Fifteen noncoding genes with an enriched mutation rate compared to surrounding nonannotated regions. P-values are derived from the Fisher exact test (two-sided) and corrected for multiple testing using the FDR (Hochberg) method.
aENSG not recognised by ICGC data portal.

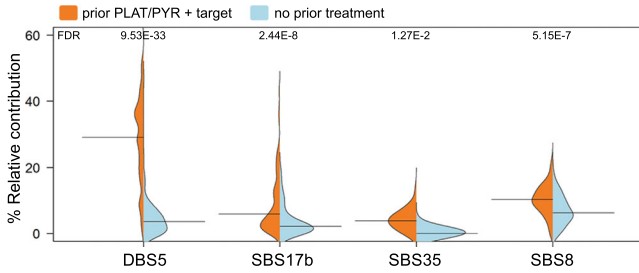

**Fig. 3 Mutational signatures in prior-treated cases compared to untreated cases.** Relative contribution (%) of several single and double base mutational signatures (SBS/DBS) in patients receiving prior treatment with platinum, pyrimidine antagonist, and targeted anti-EGFR treatment (PLAT/PYR + target; orange, $n = 134$) compared to untreated patients (blue, $n = 124$). Horizontal lines indicate the median. P-values are derived from the MWU test (two-sided) and corrected for multiple testing using the FDR (Hochberg) method. Source data are provided as a Source Data file.

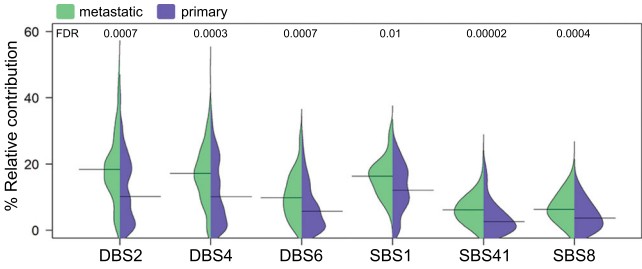

**Fig. 4 Mutational signatures in primary CRC and untreated metastatic CRC.** Relative contribution (%) of several single and double base mutational signatures (SBS/DBS) in primary CRC tumors (purple, $n = 73$)[17], compared to untreated metastatic CRC tumors (green, $n = 124$). Horizontal lines indicate the median. P-values are derived from the MWU test (two-sided) and corrected for multiple testing using the FDR (Hochberg) method. Source data are provided as a Source Data file.

characterized by higher relative contributions of signatures related to prior treatment (SBS5/8/35/17a/17b and DBS5) and signatures known from primary CRC (SBS1/5/18/40, DBS9), respectively. Samples from Cluster 6 are enriched (Fisher's exact $p = 0.005$) for samples with >5% contribution of the recently described *E. coli* mutational signature in CRC as well[21]. Cluster 3 was labeled 'mCRC-specific' as it contains both patients with ($n = 63$) and without ($n = 31$) prior treatment characterized by higher relative contributions of signatures SBS9/37/39/41, which, except for SBS37, are rarely detected in primary CRC. Etiologies for SBS37/39/41 are unknown, whereas SBS9 mutations have been partly associated with polymerase eta (Pol η) function during somatic hypermutation in lymphoid cells. In vitro, Pol η activity has been associated with anticancer drugs resistance, specifically cisplatin and 5-FU[22–24]. Indeed we find that the majority of patients (13 out of 15) in cluster 3 with a high SBS9 contribution (≥10%) had already received prior treatment, although this did not reach statistical significance (Fisher's exact test $p = 0.07$).

The remaining minor groups are found in Clusters 2, 4, and 5. Samples in clusters 2 and 4 are defined by a large contribution of DBS8 and DBS2, respectively. Cluster 5, labeled 'high TMB', contains 14 samples, which were all characterized by a high TMB (defined as >10/Mb) compared to only 82 out of the 415 remaining samples (20%) in the other clusters. High contributions of DNA mismatch repair associated signatures SBS15/44 and DBS7 characterize the 13 MSI samples in this cluster,

whereas the one remaining sample showed high contributions of SBS10a/b, associated with polymerase epsilon (*POLE*) mutations.

**MSI-specific gene mutations**. We subsequently investigated whether specific somatic gene mutations were associated with each of the six clusters described above and found this was true only for the high TMB cluster (cluster 5). To correct for the higher likelihood of finding any mutation in a high TMB sample, we applied a permutation test[25,26], which identified 28 genes as significantly more frequently mutated in the high TMB cluster versus all other samples (Fisher exact test, FDR and permutation $p < 0.05$, see Supplementary Table 4). As these 28 genes are large (cDNA size range 1.5–22 kb) and often contain substantial numbers of microsatellites and mononucleotide stretches (range 4–126), we evaluated whether their observed mutation frequency in MSI cases was significantly higher than the frequency distribution observed for all other genes with a comparable number (±10%) of MSI-prone coding sequences. Except for *TNXB*, for which we were unable to establish a reliable control distribution, all identified genes were significantly more frequently mutated in MSI cases compared to control genes containing similar numbers of MSI-prone sequences (one sample sign test; all $p \leq 0.0001$). These results suggest that mutations in these genes are selected for during the disease process in MSI tumors. The top 2 genes, *ACVR2A* and *UBR5*, are known targets of the MSI process[27]. *LRP1* mutations were found to reduce its expression in CRC and were associated with MSI status and poor outcome[28]. Although the other 25 identified genes were not previously associated with MSI status, three of these genes (*KMT2C*, *KMT2D*, and *FAT1*) were present in the Yaeger dataset of mCRC samples[6]. Mutations in all three overlapping genes were significantly enriched in MSI cases ($n = 16$) compared to microsatellite stable (MSS) cases ($n = 305$) in this dataset as well (all Fisher $p < = 9.19E-7$).

**Association between molecular landscape and treatment response**. The observed molecular characteristics were associated with response to current treatment for the 286 patients in our cohort with recorded treatment response. These results should be interpreted with caution due to the heterogeneity of our cohort in terms of both treatment line and type of prior treatments received, which may introduce bias. We studied ordinal response (PD, SD, and PR) to any treatment as well as to specific treatment regimens. In total, 123 items were used as input in the regression model, consisting of five themes (full list in Supplementary Data 3): clinical parameters (age, gender, prior treatment, and radiotherapy), counts (TMB, kataegis, chromothripsis, total number of SV by type and the number of 10kb–1Mb deletions), mutational signatures (DBS/SBS), driver genes (including non-coding genes), and GISTIC-defined CNVs. Items that reached univariate statistical significance ($p < 0.05$) were used in a multi-variable penalized ordinal regression model for treatment response (Table 4).

Overall we found that, next to receiving prior treatment(s), the number of 10kb–1Mb deletions, mutations in *KRAS*, *APC*, *PIK3CA*, and *LINC00672*, mutational signatures SBS17b/39, DBS2/5/11, and gains at 18p, 17q, and 20q were associated with treatment response regardless of treatment type in mCRC patients. For SBS17b this effect was more pronounced when specifically investigating patients treated with platinum as described before[17]. CNVs were predominantly associated with response to PLAT/PYR or PYRmono treatment, whereas mutations in *FBXW7* were associated with poor response to targeted treatment. *FBXW7* mutations were detected in 51 patients from our cohort, including 21 *KRAS* wild-type patients. Of these 21 patients, five were treated with panitumumab

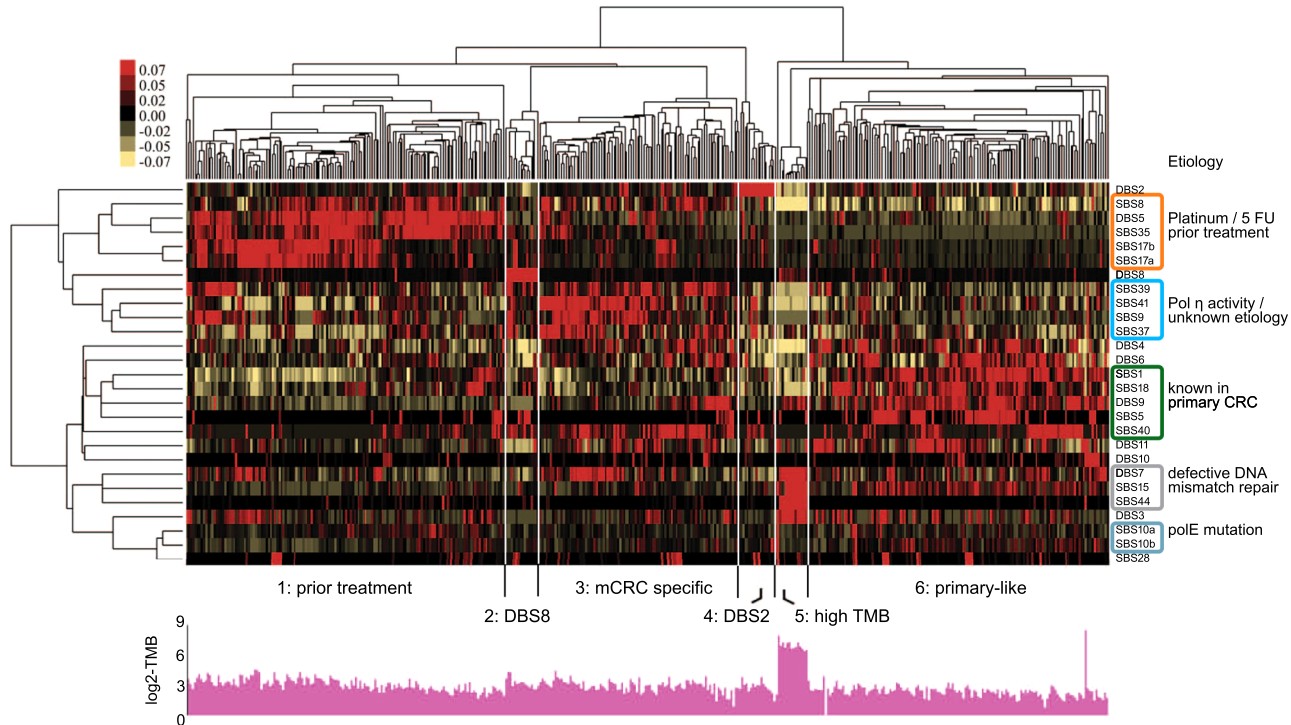

**Fig. 5 Unsupervised hierarchical clustering of metastatic CRC using relative contribution of preselected mutational signatures.** Heatmap representing the median-centered relative contribution of mutational signatures between samples. Values were scaled from red (relative contribution above median) to yellow (relative contribution below median). Included single and doublet base signatures (SBS/DBS) are indicated at the right to which etiologies are added when known. Grouping of samples is shown by the dendrogram at the top. Source data are provided as a Source Data file.

monotherapy, all of whom had PD as best response. This suggests that, next to somatic *KRAS* mutations, somatic *FBXW7* mutations may provide an additional negative selection marker for anti-EGFR treatment. This finding is in concordance with previous reports on *FBXW7* mutation prevalence in nonresponding patients on anti-EGFR treatment[29,30].

**Potential clinical implications**. WGS data of our cohort of 429 patients with metastatic CRC revealed several potential molecular features that might be associated with sensitivity to particular anticancer agents. A high TMB (here defined as >10 mutations per Mb) has been suggested as a potential selection tool for tumors that may respond to immunotherapy[31]. In our cohort, 96 (22%) samples showed a TMB > 10, of which 13 were MSI. A gradual increase in TMB was observed with the number of prior treatments (test for trend, $p = 4.39E\text{-}13$). For the subset of samples of which we also had RNA-seq data available, we calculated the Tumor Infiltrating Leukocyte (TIL) score as a proxy for the immunogenicity of the tumor[32]. Interestingly, we did not observe a significantly higher TIL score in the TMB-high samples ($n = 21$) compared to the other samples ($n = 63$; MWU; $p = 0.39$), whereas the average TIL score in MSI samples is significantly higher compared to both MSS samples with a high TMB and with a low TMB (Kruskal–Wallis test ($p = 0.037$) followed by Dunn's pairwise comparison (Benjamini–Hochberg corrected $p = 0.012$ and $p = 0.021$ for MSI compared to MSS with high and low TMB, respectively (See Supplementary Fig. 2). Although far from definite, these results support the on-label use of immunotherapy in MSI tumors and suggest that merely using TMB may not be sufficient to identify the tumors with immunogenic potential in the metastatic setting.

Other on-label markers found in our cohort include a targetable *BRAF* V600E mutation in 40 patients, as well as 130 *RAS/RAF* wild-type patients that did not receive targeted anti-

EGFR treatment yet. However, our data suggest that mutations in *FBXW7*, observed in 21 out of these 130 *RAS/RAF* wild-type patients, should be considered as a contra-indication for the use of anti-EGFR treatment. Molecular biomarkers for potential off-label use that were found in our cohort include amplifications of *ERBB2* (HER2), *MET* and *CDK4*, loss of *BRCA1* and *BRCA2* through deletion or high impact mutations, loss of *TSC1* and *TSC2* through high impact mutations, and possible fusions of *PDGFRB*. In addition, 23 patients in our cohort carried a *KRAS* G12C mutation, for which an inhibitor may become available in the near future[33].

In summary, for 55% of our patients one or more targeted treatments are potentially available based on the molecular profile of their cancer (Fig. 6).

**Discussion**

This study encompasses a WGS-based, comprehensive description of the molecular landscape of metastatic CRC and aims to put this landscape into perspective by associating it with prior systemic treatments, comparing it to primary CRC and relating it to treatment response.

In general, the genomic landscape of CRC remains relatively stable in metastatic disease. However, compared to primary CRC, our metastatic CRC cohort showed significant enrichment for mutations in 4 out of 23 coding and 12 out of 15 noncoding (putative) driver genes. From the identified putative drivers, only mutations in *PIK3CA* were significantly decreased in mCRC. Six of our identified coding driver genes are not present in the current CRC-specific MSK-IMPACT panel, namely *ZFP36L2*, *BCL*, *BCL9L*, *ELF3*, *LMTK3*, and *TGIF1*.

Within the mCRC cohort we observed clear effects of received prior treatments on the total numbers of aberrations, CNVs, and mutational signatures, with the latter sufficiently dominant to show up as a separate group after hierarchical clustering.

**Table 4 Multivariate LASSO analysis.**

| Type | Item | | All treatments | Oxaliplatin containing | PLAT/PYR | Target-mono | TOP/PYR |
|---|---|---|---|---|---|---|---|
| Clinical | Prior Treatment | | 0.57 | | | | |
| | Gender | | | 0.77 | | −1.14 | −1.10 |
| Counts | nr of Tandem Duplications | | | | | 1.19 | |
| | nr of 10kb–1Mb deletions | | 0.01 | | | | |
| Mutational | DBS2 | | −0.02 | | | | |
| Signatures | DBS5 | | 0.13 | | | | |
| | DBS11 | | −0.03 | | | | |
| | SBS17b | | 0.07 | 0.13 | | | |
| | SBS39 | | 0.04 | | | | |
| | SBS41 | | | | −0.21 | | |
| Driver Genes | APC | | 0.23 | | | | |
| | KRAS | | 0.78 | | | | |
| | PIK3CA | | 0.22 | | | | |
| | FBXW7 | | | | | 17.65 | |
| Non-coding | LINC00672 | | 0.90 | | | | |
| GISTIC Regions | Gain 17q12 | (ERBB2*) | 0.44 | | | | |
| | Gain 18p11.32 | (CETN1*) | 0.59 | | | | |
| | Gain 20q11.1 | (BCL2L1*) | 0.12 | | | | |
| | Gain 8p11.21 | (KAT6A) | | −0.78 | −1.56 | | |
| | Gain 7p21.3 | (VWDE) | | | | | 3.68 |
| | Gain 7q31.2 | (MET*) | | | | | 3.59 |
| | Gain 7p12.3 | (PKD1L1) | | | | | 3.04 |
| | Gain 7q34 | (no genes in peak) | | | | | 3.59 |
| | Gain 14q23.1 | (no genes in peak) | | | | | 1.64 |
| | Loss 18q12.2 | (hsa-mir-924*) | | −1.52 | −3.30 | 2.79 | |
| | Loss 6q26 | **(PARK2)** | | | −1.38 | | |
| | Loss 9p21.3 | **(CDKN2A*)** | | | −1.88 | | |
| | Loss 16q23.1 | **(WWOX)** | | | −1.70 | | |
| | Loss 4q22.1 | **(CCSER1)** | | | | 2.66 | |
| | Loss 4q35.1 | (IRF2) | | | | 1.78 | |
| | Loss 18q21.2 | (SMAD4) | | −1.35 | −3.30 | 2.79 | |
| | Loss 18q23 | (NFATC1*) | | −1.14 | −3.30 | | |
| | Loss Xp22.31 | (STS*) | | | | −1.40 | 1.78 |
| | Loss 14q23.3 | **(GPHN)** | | | | | −2.09 |

Items that reached univariate statistical significance ($p < 0.05$) were used in a multivariable penalized ordinal regression model for treatment response. Univariate regression was performed for genomic features (Supplementary Data 3) using the 'polr' function from the MASS R package (v7.3-51.4) and subsequently those with a univariate two-sided $p$-value <0.05 were selected for multivariable ordered LASSO regression using the ordinalNet R package (v2.7). Regression coefficients are shown for features that remained significant in the multivariable model.
*Multiple genes present in region; bold: known Fragile Site region

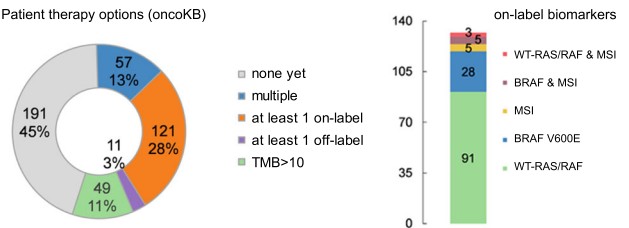

Patient therapy options (oncoKB)

- none yet — 191 / 45%
- multiple — 57 / 13%
- at least 1 on-label — 121 / 28%
- at least 1 off-label — 11 / 3%
- TMB>10 — 49 / 11%

on-label biomarkers

- WT-RAS/RAF & MSI — 3
- BRAF & MSI — 5
- MSI — 28
- BRAF V600E — 91
- WT-RAS/RAF

**Fig. 6 Actionable genes.** Data from OncoKB were matched to affected genes observed in our mCRC cohort. Numbers indicate the number (and percentage) of affected patients. Source data are provided as a Source Data file.

Remarkably, we also observed an mCRC-specific cluster characterized by signatures which are rarely found in primary CRC and are not associated with any treatment (SBS9/39/41). SBS9 is associated with Pol η activity, an error-prone polymerase encoded by the *POLH* gene, which mediates translesion synthesis and is induced by replication stress[34]. Interestingly, high levels of Pol η have been associated with cancer therapy resistance in vitro[22–24]. We did observe that the majority of patients with a high relative SBS9 contribution had already received prior treatment; however, unfortunately, sample numbers were too low to directly associate

SBS9 contribution with *POLH* expression in our dataset. Another predominant cluster group consisted of metastatic MSI samples. In these samples we observed a significant enrichment of mutations in a specific set of genes compared to other similarly MSI-prone genes, suggesting these genes are preferentially affected or selected for during disease progression.

The varying number and types of received prior treatments within our cohort hampered the search for prognostic and predictive biomarkers. However, we found that, next to already known events, the number of *LINC00672* mutations and 10kb–1Mb deletions were associated with treatment response irrespective of the type of treatment. Strikingly, many of these recurrent deletions occur in known Common Fragile Site (CFS) genes, as described in primary CRC as well[13], implicating replication stress as one of driving mechanisms[35]. In addition, *FBXW7* mutations were predictive for poor response to EGFR-targeted treatments in our prospective cohort. This is in line with previous observations showing that *FBXW7* mutations were enriched in unresponsive patients compared to patients responding well to EGFR-targeted treatments[29,30].

The current study gives a detailed description of the genomic landscape of metastatic CRC. More specifically, our study identifies treatment-induced changes, metastasis-specific alterations, and associations between molecular traits and response to treatment. In addition, we provide prospective validation for *FBXW7*

mutations as a predictive biomarker for poor response to EGFR-targeted treatment. Combined with future studies, this catalogue of molecular alterations will speed up the identification of resistance mechanisms, the determination of metastasis-driving processes, and, ultimately, the improvement of metastatic CRC patient care.

## Methods

**Patient cohort and study procedures.** Colorectal cancer patients included in this study were selected from the previously described cohort of the Center for Personalized Cancer Treatment (CPCT) consortium (CPCT-02 Biopsy Protocol, ClinicalTrial.gov no. NCT01855477), which was approved by the medical ethics committee of the University Medical Center Utrecht, the Netherlands[9]. All patients have given explicit consent for whole-genome sequencing and data sharing for cancer research purposes. Upon our data request for all CRC patients thus far, we were provided with the data of all patients registered as metastatic CRC patients included between April 2016 and January 2019 ($n = 487$). Patients who received systemic treatment which is not normally given to colorectal cancer patients (e.g., carboplatin, paclitaxel, sunitinib, and etoposide) were excluded to avoid erroneous inclusion of patients suffering from another type of cancer ($n = 28$). When multiple biopsies were included for one patient ($n = 29$), only the first biopsy was included in our analyses. In total, we included 429 distinct CRC patients in our analyses. Based on the provided information regarding all forms of systemic treatment patients received before the study biopsy took place (further referred to as "prior treatment"), we coded the (groups of) active agents using the following abbreviations: PLAT (oxaliplatin), PYR (fluoropyrimidines), TOP (topoisomerase inhibitor; Irinotecan), +targeted (when bevacizumab or panitumumab was added), CHEMCOM (triplet combination therapy). Prior-treatment regimens were grouped based on their working mechanism to enable the analysis of their effect on the genomic landscape. Treatment related analyses were performed using combinations of the abbreviations mentioned above. For detailed information see Table 1.

**Whole-genome sequencing; identification of somatic changes.** Whole-genome sequencing of paired tumor/normal was performed in all cases. In short, raw sequencing data were processed using bcl2fastq (versions 2.17 to 2.20), mapped to the human reference genome GRCh37 using BWA-mem v0.7.5a and GATK BQSR and Haplotype Caller v3.4.46 and Strelka v1.0.14 were used to call somatic mutations. Within our cohort, 98% of the biopsies of metastatic lesions showed a coverage of at least 30× (95% with >60× coverage), whereas for the normal blood 98% had >10× coverage and 94% >20× coverage. The identification of copy number changes was performed using GISTIC v2.0.23[36] with the following parameters: genegistic 1; gcm extreme; maxseg 4000; broad 1; brlen 0.98; conf 0.95; rx 0; cap 3; saveseg 0; armpeel 1; smallmem 0; res 0.01; ta 0.1; td 0.1; savedata 0; savegene 1; and qvt 0.1[9,20].

**RNA sequencing and CMS calling.** Matched RNA was isolated from the same frozen tissue for 91 CRC patients on an automated setup (QiaSymphony) according to supplier's protocols (Qiagen) using the QIAsymphony RNA Kit for tissue and quantified by Qubit. A total of 50–100 ng of RNA was used as input for library preparation using the KAPA RNA HyperPrep Kit with RiboErase (Human/Mouse/Rat) (Roche). Barcoded libraries were equimolarly pooled and sequenced using standard settings (Illumina) on either a NextSeq 500 (V2.5 reagents) generating 2 × 75 read pairs or a NovaSeq 6000 generating 2 × 150 read pairs. BCL output was converted to FASTQ using bcl2fastq (versions 2.17–2.20) using default parameters and sequence reads were trimmed for adapter sequences using fastp (v0.20.0). The resulting FASTQ files were mapped to GRCh38 using STAR (v2.6.1d)[37]. Sambamba (v0.7.0)[38] was used to mark duplicates and index the resulting BAM files. Gene annotation was derived from GENCODE Release 30 (https://www.gencodegenes.org/), raw read counts were obtained with feature-Counts (v1.6.3)[39] and normalized using GeTMM[40]. Normalized data were used to (1) determine CMS with both the single-sample prediction parameter from the "CMSclassifier" package (v1.0.0) (https://github.com/Sage-Bionetworks/CMSclassifier)[2] and CMSCaller v(0.99.1)[11], and (2) calculate the Tumor Infiltrating Lymphocytes (TIL) score by averaging the expression of TIL-genes[32].

**Identification of mutational signatures and driver genes.** Mutational signatures (COSMIC v3)[17] were called using R package MutationalPatterns v1.10.0[18], focusing on single and double base signatures. This package was also used to perform de novo signature calling using the Non-negative Matrix Factorization (NMF) method. Detection of kataegis and chromothripsis was performed as previously described[41]. In short, to call kataegis only SNVs were considered to establish segments based on the intermutational distance. Segments were determined using a piecewise constant fitting model and were called as kataegis when at least five SNVs were present showing an intermutational distance of ≤2 kb. Chromothripsis-like events were called using the Shatterseek R package (v0.4). Driver genes, i.e., genes under selective pressure, were identified by the dN/dS model using R package dndscv (v0.0.1.0)[42]. A global $q \leq 0.05$ was used to select statistically significant driver genes. The R package discover v0.9.2[43] was used to test for mutual exclusivity. To identify noncoding genes with an enriched mutation

rate, we first established a baseline mutation rate based on all identified SNVs, MNVs and Indels found in nonannotated regions, as we assume these regions are not under any selective pressure. Nonannotated regions were based on GENCODE annotation (version33) and for each of these regions we calculated a mutation rate (number of mutations/size of region). Next, a mutation rate (number of mutations/size of noncoding gene) was calculated for all somatic mutations annotated as 'noncoding transcript variant'. The signed-rank test with Hochberg's multiple testing correction was used to determine whether the mutation rate of a recurrent noncoding gene (mutated in at least 5% of the cohort) exceeded the baseline mutation rate. Per noncoding gene a specific baseline was determined using only nonannotated regions (>1 kb) in an area of 2 Mb surrounding the respective noncoding gene.

**Verification in publicly available datasets.** To compare mutational signatures, publicly available WGS data from 73 primary colorectal cases were used[17]. We downloaded the matrix of counts for single and double base substitutions of primary cases and analyzed these in the same manner as the metastatic CRC cases to call mutational signatures. Observed frequencies of mutated genes in metastatic CRC were verified and compared in two publicly available datasets. Dataset 1, the Yaeger dataset, contained 321 unique metastatic CRC patients that were profiled for mutations by targeted sequencing[6]. Dataset 2, the TCGA-DFCI dataset, contained 1949 unique primary CRC patients that were profiled for mutations in coding regions (accessed via cBioPortal January 21, 2020). Prior to analysis, synonymous mutations were removed and multiple mutations within the same gene were aggregated per patient. Dataset 3, the ICGC dataset, was used to compare mutation frequencies of noncoding genes and contained 866 unique primary CRC patients with available mutation data (accessed via the ICGC data portal, release 28). The used cohorts are summarized in Supplementary Table 1.

**Estimating MSI-prone sequences.** To evaluate preferentially mutated genes in MSI cases, the number of MSI-prone sequences in a gene are of interest. Data of the Microsatellite Database (MSDB, https://data.ccmb.res.in/msdb/, June 2, 2020) were filtered for repeats annotated to human exons[44]. For each gene, the number of repeats was summed. In addition, a custom Perl script was used to count mononucleotide stretches of lengths between 6 and 13 (the latter is the minimum length used in MSDB) as we noticed many InDels in mononucleotide stretches less than 13 bases long. Exon sequences of the Consensus CDS database (https://www.ncbi.nlm.nih.gov/CCDS/) were used to count the number of mononucleotide stretches per gene.

**Associations with response to treatment.** Treatment response was evaluated according to RECIST (v.1.1) every 8 to 12 weeks depending on the treatment regimen and was defined as response (partial or complete), stable disease, or progressive disease[45]. For regression analyses, the best overall response was used as outcome measure. Genomic features (at least 5 events per group) were associated with response to treatment in a 2-step procedure using ordinal LASSO (least absolute shrinkage and selection operator) regression, which is suited for datasets with a relatively high number of predictors in comparison to cases and protects against overfitting. First, univariate regression was performed for genomic features (Supplementary Data 3) using the 'polr' function from the MASS R package (v7.3–51.4) and subsequently those with a univariate p-value <0.05 were selected for multivariable ordered LASSO regression using the ordinalNet R package (v2.7).

**Identification of potentially actionable events.** OncoKB (accessed on March 31, 2020) was used to identify clinically actionable genes from the list of mutated genes in our cohort, using only genes with level 1 and 2 evidence[46]. In case OncoKB listed a specific gene alteration as actionable genomic aberration, we only counted patients that harbored that specific mutation or CNV. For genes for which only 'Oncogenic mutations' were listed by OncoKB, we only included patients if the gene had a mutation with 'High impact' consequence (i.e., a nonsense or frameshift mutation). To evaluate patients eligible for an anti-EGFR therapy, we included only patients that were triple wild-type for KRAS, NRAS, and BRAF, and excluded those patients that had already received anti-EGFR therapy prior to biopsy.

**Statistics.** In general, a Pearson's Chi-squared test or Fisher's exact test (in case of too few expected events) was used to evaluate the categorical data while continuous variables were evaluated using either a Mann–Whitney U test (MWU) or a Kruskal–Wallis H (KWH) test depending on the number of categories. All statistical tests were two-sided and considered statistically significant when $P < 0.05$. Stata 13.0 (StataCorp) and R (v3.6.0) were used for the statistical analyses. Multiple testing using the Hochberg procedure to correct $P$ values was applied when necessary. The statistical test used is specified throughout the results section.

**Reporting summary.** Further information on research design is available in the Nature Research Reporting Summary linked to this article.

## Data availability

The WGS, RNA-seq, and corresponding clinical data used in this study was made available by the Hartwig Medical Foundation (Dutch nonprofit biobank organization)

after signing a license agreement stating data cannot be made publicly available via third party organizations. Therefore, the data are available under restricted access and can be requested upon by contacting the Hartwig Medical Foundation (https://www.hartwigmedicalfoundation.nl/applying-for-data/) under the accession code DR-058. Publicly available datasets that were used in this study are listed in Supplementary Table 1. The Yaeger data used in this study are available in the cBioPortal for Cancer Genomics (http://www.cbioportal.org/study?id=crc_msk_2017). The TCGA-DFCI data used in this study have been deposited in the cBioPortal for Cancer Genomics which we accessed on January 21, 2020 (https://www.cbioportal.org/study/summary?id=coadread_tcga; https://www.cbioportal.org/study/summary?id=coadread_tcga_pub; https://www.cbioportal.org/study/summary?id=coadread_tcga_pan_can_atlas_2018; https://www.cbioportal.org/study/summary?id=coadread_dfci_2016). The ICGC data used in this study have been deposited in the ICGC data portal (release 28) (https://dcc.icgc.org/projects/COAD-US; https://dcc.icgc.org/projects/COCA-CN; https://dcc.icgc.org/projects/READ-US). The remaining data are available within the Article, Supplementary Information or available from the authors upon request. Source data are provided with this paper.

## Code availability

The bioinformatical code used for data processing is available at https://github.com/hartwigmedical/pipeline5.

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

## Acknowledgements

We thank the Hartwig Medical Foundation, and Stichting Stelvio for Life for financial support of clinical studies and WGS analyses. We thank the Center for Personalized Cancer Treatment for proving the clinical data. We would like to thank all local principal investigators, medical specialists, and nurses of all contributing centers for their help with patient accrual. We are particularly grateful to all participating patients and their families.

## Author contributions

P.A.J.M., M.S., S.S., J.W.M.M., and S.M.W. wrote the manuscript, which all authors reviewed. M.S. and J.V.R. performed the bioinformatics analyses. P.A.J.M., L.A., and S.S. managed clinical data assessment. M.L., N.S., M.P.H., G.A.C., J.M.V.R., and A.J.T.T. are main clinical contributors. M.P.L. and S.S. are members of the CPCT-02 study team and/or CPCT board. E.C. coordinated the sequencing of samples and contributed to the bioinformatics analyses.

## Competing interests

The authors declare no competing interests.
