## [Peer Review File · Nature Communications]

REVIEWER COMMENTS

Reviewer #1 (Remarks to the Author): expert in CRC precision medicine and clinical trials

In their manuscript, Mendelaar et al. have reported a whole genome sequencing (WGS) analysis of metastases in a cohort of 429 CRC patients enrolled in the CPCT-02 Biopsy study (NCT01855477). CPCT is a sample procurement clinical platform in metastatic solid tumor patients combining histological biopsy of tumor material with NGS to improve the selection of patients for targeted therapy trials. In addition to DNA sequencing, one fifth of cases also underwent transcriptional profiling by RNA-Seq for CMS classification. Cases were divided in order to observe the effects of prior treatment on genome landscape. The authors have reported an increase in terms of genomic TMB, number of SVs, and number of affected CNV regions, as well as altered relative contributions for several mutational signatures in defined pre-treatment groups compared to treatment-naïve patients. Differences in mutational signature contributions and driver genes between CRC metastases and primary tumor (already published data) were also reported.

The manuscript is well written, the statistical aspects are correct and the study could have a potential clinical value since 55% of cases were found to have an actionable target (i.e. a target for which a drug is available, albeit not necessarily approved for mCRC). However, my concern is on the novelty and suitability for publication in Nature Communication. Indeed, despite a tremendous data generating effort, results are descriptive and not followed-up by further analyses a/o experiments that could improve the translational interest of the results and functionally validate them for future deployment in the clinic.

Three examples (but there could be more):

1. In general: quoting the authors, beside allowing for “ [a] more reliable evaluation of mutational signatures and copy number alterations than WES” it is not at all clear what is the added value of a WGS-based approach in mCRC. Explaining why WGS is more informative than WES or CRC-specific NGS panels, is pivotal to leverage the potential clinical value of a resource-cumbersome approach such WGS and the translational interest of the manuscript. Curiously, the authors are underplaying this pivotal aspect to such an extent to include in the discussion the following rather surprising sentence “ [WGS] ..In addition, raises the opportunity to investigate the non-coding part of the genome. However, as this was already studied in detail across multiple cancer types including most of our current cohort we have not addressed this in the current study”.
2. Main findings: authors report as a major achievement the finding that mutations of the FBXW7 gene seems to correlate (in a retrospective analysis) to primary resistance to anti-EGFR therapies. This is actually old news, see Lupini et al., BMC Cancer 2015; 15:808 doi: 10.1186/s12885-015-1752-5 and Korphaisarn K, et al Oncotarget. 2017. doi:10.18632/oncotarget.16848, neither of which are cited in the manuscript. What one would expect in a Nat Comm paper is (at the very least) the in vitro preclinical validation of FBXW7 mutation(s) as predictor(s) of response to cetuximab/panitumumab plus, a set of experiments to pinpoint the mechanistic reasons of such a finding. Especially given that FBXW7 is a critical tumor suppressor and one of the most commonly deregulated ubiquitin-proteasome system proteins controlling the degradation of such ‘oncoplayers’ as cyclin E, c-Myc, Mcl-1, mTOR, Jun, Notch etc.
3. Other findings (again as examples): i) one of the most interesting finding in my opinion is that one

of the metastases specific signatures (BS9) is partly associated to the error-prone polymerase. This finding, once again, is reported almost en passant and is not functionally validated by an experiments ii) Comparison and validation of TIL score and IHC in 6 MSI samples vs 6 MSS high TMB vs 6 MSS low TMB should be shown in the manuscript.

Reviewer #2 (Remarks to the Author): expert in mutational signatures and mutagenesis

In this paper, Wilting and colleagues report WGS of 429 pairs of primary CRC tumours and metastases. There is also transcriptomic data for a subset of these samples. They ran a number of analyses, including correlation with prior treatment of the primary CRC. The stated aim of this study is "to provide a comprehensive description of the molecular landscape of metastatic CRC".

Overall, this is an interesting paper that has yielded some new insights. I have some suggestions for improvements and requests for clarifications.

1. The paper's title is rather vague, it can be improved by being more specific about key finding(s).
2. Throughout the paper, the terms "pretreatment" or "prior treatment" are used interchangeably. But they aren't defined explicitly, as far as I can see, so it took a while to know for sure that this refers to treatment of the primary CRC. Clearly stating this at the beginning would help.
3. "Patients with treatment data not matching standard of care...were excluded..." That's rather nebulous. Please elaborate on what this means specifically?
4. With respect to calling variants from WGS, what is the minimum allele frequency to be called? To what extent are subclonal variants being investigated?
5. 29% of the CRC samples were not classified into a CMS category. Are these somehow intermediate between categories?
6. I wonder about rearrangement breakpoints, it seems they should be investigated as part of a comprehensive description?
7. As far as I can see, de novo NMF mutational signature extract was not done on the WGS data. I think it would be worthwhile to do this, to independently verify prior signatures associated with CRC and to check if there might be other novel signatures mixed in with the known prior signatures.
8. "Remarkably, no mutations associated with specific pretreatments or with pretreatment in general were found." This means that mutations in the primary CRC samples were a subset of the mutations observed in the mets, correct? That's not necessarily a surprising result, considering the

metastases are genetically descended from the primaries, or am I missing something?

9. The phrase "no non-driver genes with an increased frequency" on line 275 is hard to follow. A reader can figure out what is meant by looking at the table, but it's better to rewrite that long sentence and split it up into two sentences.

10. To what extent are TILs contributing to the SBS9 observed in cluster 3?

11. For Figure 5, what are the cutoffs for higher and lower % contribution? Why not use a continuous colour spectrum instead of imposing arbitrary cutoffs?

12. Are there distinguishing features for the set of 28 "MSI genes"? GO enrichment? Enrichment of protein domains, DNA motifs?

13. It seems the analysis of association between molecular features and treatment results would be a suitable problem for machine learning. The authors should try using some off the shelf ML algorithms and compare them to their current results.

14. The sentence at lines 403-405 is really confusing and should be rewritten.

Reviewer #3 (Remarks to the Author): expert in CRC genomics

Congratulations to Mendelaar for putting together promising study results. The manuscript is technically well written, but major re-analysis of the data is recommended (and consequently formatting of results) for publication in Nature Communications. It is hard for me to identify take-home messages. WGS and RNAseq in over 400 clinically annotated metastatic CRC cases and someone would expect more "discoveries". It may be possible, given that genomics is relatively stable over time (and RNAseq was not explored in depth). My suggestions are:

1. in terms of methodology, CMS classification with CMSclassifier in metastatic tissues is not accurate. You may use CMScaller, which uses mostly cancer cell intrinsic signals (microenvironment of metastases are different).

2. explore additional immune cell infiltration signatures from RNAseq, beyond TILs, and compare with patterns seen in primary CRC.

3. the mutational signatures results are difficult to interpret. Any relevant association beyond primary versus metastasis, anti-EGFR exposure? Does number of treatment lines impact shifts in mutation signatures? Have you looked at the impact of different signatures on survival in the metastatic setting?

4. the clinical correlates of the study are very poor. Theoretical actionability is not insightful (very few places in the world can execute WGS for clinical decision). I suspect your pragmatic actionability is very low. So my suggestion is to explore the added value of WGS over targeted panels (Foundation Medicine-like)? A negative result is acceptable. Maybe look at the added value of metastatic tissue NGS at later stages over primary NGS. In addition, I suggest you look more carefully at other endpoints - super responders to anti-EGFR therapy versus non-responders? Any other outlier clinical

scenario where deep
molecular profiling may add value?

Response to the referees

Reviewer 1

In their manuscript, Mendelaar et al. have reported a whole genome sequencing (WGS) analysis of metastases in a cohort of 429 CRC patients enrolled in the CPCT-02 Biopsy study (NCT01855477). CPCT is a sample procurement clinical platform in metastatic solid tumor patients combining histological biopsy of tumor material with NGS to improve the selection of patients for targeted therapy trials. In addition to DNA sequencing, one fifth of cases also underwent transcriptional profiling by RNA-Seq for CMS classification. Cases were divided in order to observe the effects of prior treatment on genome landscape. The authors have reported an increase in terms of genomic TMB, number of SVs, and number of affected CNV regions, as well as altered relative contributions for several mutational signatures in defined pre-treatment groups compared to treatment-naïve patients. Differences in mutational signature contributions and driver genes between CRC metastases and primary tumor (already published data) were also reported.

The manuscript is well written, the statistical aspects are correct and the study could have a potential clinical value since 55% of cases were found to have an actionable target (i.e. a target for which a drug is available, albeit not necessarily approved for mCRC). However, my concern is on the novelty and suitability for publication in Nature Communication. Indeed, despite a tremendous data generating effort, *results are descriptive and not followed-up by further analyses a/o experiments that could improve the translational interest of the results and functionally validate them for future deployment in the clinic.*

Response: Although we do agree that our manuscript is a descriptive one, we would like to emphasize that this is a crucial first step towards translation of genomics into the clinic. The current study provides an unprecedented detailed description of the genomic landscape of metastatic lesions from colorectal cancer and illustrates the potential direct clinical value of using genomics in metastatic cancer patients. In our opinion, only after studies like these, follow-up prospective studies can be designed providing evidence (or not) of clinical utility of the targets revealed by WGS. As such, this study greatly contributes to paving the way for increased use of genomics in the clinical management of metastatic colorectal cancer patients.

Comment 1a

It is not at all clear what is the added value of a WGS-based approach in mCRC. Explaining why WGS is more informative than WES or CRC-specific NGS panels, is pivotal to leverage the potential clinical value of a resource-cumbersome approach such WGS and the translational interest of the manuscript.

Response: The use of WGS allowed us to investigate mutations (including the identification of novel driver genes) not included in the current CRC-specific MSK-IMPACT panel, such as ZFP36L2, BCL, BCL9L, ELF3, LMTK3 and TGIF1), mutational signatures, MSI status (diagnostically very relevant in CRC), genome-wide chromosomal copy number changes, and structural rearrangements in one single experiment, thereby generating a detailed and unbiased genomic landscape of mCRC. All these genomics analyses are unique to WGS that are not possible (or much less reliable so) using WES or panels. The added value lies in providing a foundation to scout for candidates that can be used in targeted approaches should this disease prove homogeneous enough to treat patients more or less

uniformly based on the genetic evidence. If too heterogeneous, personalized medicine may benefit from WGS as a one-shot diagnostic test, providing all necessary knowledge to optimally treat patients. To further substantiate the value of WGS we have now also added results on recurrent mutations in the non-coding part of the genome, as well as the occurrence of chromothripsis and kataegis to the manuscript:

Introduction [lines 31-40]

To date, large in-depth analyses of colorectal cancer metastases are limited to studies using either whole exome sequencing (WES) or targeted sequencing of cancer-associated genes^{4, 5, 6}. Although these studies yielded extensive knowledge on the presence of specific genomic aberrations in mCRC, they do not necessarily reflect its complete molecular landscape. For optimal identification of mutational signatures, the power provided by WGS data greatly exceeds that of WES⁷. Next to this, WGS simultaneously allows for the determination of MSI, structural rearrangements, chromothripsis, and kataegis. In addition, clinically relevant genetic alterations within non-coding regions were recently reported in primary CRC⁸. To date, the only other study which reported in detail on whole genome sequencing (WGS) data of colorectal metastases included 12 patients⁴.

Results - The molecular landscape of mCRC [lines 81-85]

Chromothripsis was observed in 47 cases (11%), whereas kataegis was observed in 102 cases (24%), involving just a single chromosomal region in two-third of cases, with a maximum of 10 regions in one single case. Presence of kataegis was associated with MSI and high TMB (≥ 10 ; test for trend $p=0.00014$). In fact, 9 out of 13 MSI cases had at least 2 kataegis regions.

Results - Effects of systemic prior treatment on the genomic landscape [lines 118-121]

Patients having received prior systemic treatment ($n = 284$) showed a significantly higher TMB, number of SVs, number of affected GISTIC CNV regions (7.58 versus 5.82; 208 versus 148; 31 versus 28, respectively; MWU p -values < 0.005), and occurrence of chromothripsis (6.5% versus 13.4%; Fisher exact test $p=0.042$) compared to those ($n = 124$) who did not.

Results: The molecular landscape of mCRC [lines 103-107]:

Similarly, for 15 non-coding genes an enriched mutation rate was observed compared to surrounding non-annotated regions (Table 3), suggesting these genes are relevant for the oncogenic process. These non-coding genes include *PTENP1*, a known tumor suppressor in CRC¹⁴, *MALAT1*, for which an increased mutation rate was already described in a pan-cancer analysis¹⁵, and *LINC00672*, described to promote chemo-sensitivity¹⁶.

Results: Comparing metastatic CRC to primary CRC [lines 158-160]:

With respect to the identified putative non-coding drivers (Table 3), except PIPSL and PTENP1 all of them were enriched in mCRC compared to primary CRC (ICGC dataset; Fisher exact test, $FDR < 5.74E-4$).

Comment 1b

Curiously, the authors are underplaying this pivotal aspect to such an extent to include in the discussion the following rather surprising sentence " [WGS] ..In addition, raises the opportunity to investigate the

non-coding part of the genome. However, as this was already studied in detail across multiple cancer types including most of our current cohort we have not addressed this in the current study”.

We agree on the contradictory character of this statement. We did study the non-coding regions in some respects, but did not mention our results in the manuscript previously because we wanted to focus on our more clinically targetable results. We have now greatly extended our non-coding analyses, using a custom-made method to address this properly. Mutations in non-coding genes were compared to the mutation rate observed in non-annotated regions (regions where no known gene is annotated). In total, 26 non-coding genes had at least 22 mutations, of which 15 showed a significantly enriched mutation rate (signed-rank test, FDR corrected $p < 0.05$), indicating they represent potential driver genes (see table below). Almost all non-coding genes were more frequently affected in mCRC compared to primary CRC and LINC00672 mutations associated with response to therapy in a multivariate setting.

ENSG	Symbol	Size	Chr	type	mutation rate	FDR Hochberg	Metastatic CRC		primary CRC, ICGC		Fisher p-value	FDR Hochberg
							Mutations (N)	Mutations (%)	Mutations (N)	Mutations (%)		
ENSG00000273001	AL731533.2	577	10	lncRNA	0.067591	0	6	1.4	0	0	0.001291	3.87E-03
ENSG00000280325	AC074183.2	921	7	TEC	0.033659	0	25	5.8	*	*		
ENSG00000261584	AL513548.1	1723	6	lncRNA	0.012768	9.93E-24	14	3.3	3	0.3	3.84E-05	2.09E-04
ENSG00000259834	AL365361.1	3480	1	lncRNA	0.007184	2.48E-09	17	4.0	0	0	5.62E-09	7.30E-08
ENSG00000264920	AC018521.5	4583	17	lncRNA	0.00851	1.02E-08	16	3.7	1	0.1	2.06E-07	1.85E-06
ENSG00000231784	DBIL5P	2676	17	transcribed_	0.008595	6.26E-07	18	4.2	1	0.1	2.39E-08	2.63E-07
				unitary_pseudogene								
ENSG00000273033	LINC02035	5475	3	lncRNA	0.006758	9.54E-06	23	5.4	0	0.0	6.15E-12	8.61E-11
ENSG00000266979	LINC01180	3985	17	lncRNA	0.005521	1.18E-05	13	3.0	1	0.1	5.00E-06	4.00E-05
ENSG00000272070	AC005618.1	3147	5	lncRNA	0.006991	4.19E-05	18	4.2	1	0.1	2.39E-08	2.63E-07
ENSG00000251562	MALAT1	8828	11	lncRNA	0.005551	0.000283	30	7.0	19	2.2	4.19E-05	2.09E-04
ENSG00000261094	AC007066.2	2710	9	lncRNA	0.008118	0.000287	11	2.6	2	0.2	1.86E-04	7.43E-04
ENSG00000263874	LINC00672	4216	17	protein_coding	0.005455	0.000493	14	3.3	2	0.2	9.70E-06	6.79E-05
ENSG00000180764	PIPSL	3349	10	transcribed_	0.006569	0.0013	14	3.3	34	3.9	0.640	0.640
				processed_pseudogene								
ENSG00000237984	PTENP1	3995	9	transcribed_	0.005507	0.0013	15	3.5	22	2.5	0.376	0.640
				processed_pseudogene								
ENSG00000240859	AC093627.4	5916	7	lncRNA	0.006085	0.0048	18	4.2	2	0.2	1.66E-07	1.66E-06

Commentary figure 1, table 3 in manuscript

We agree that reporting these results will enhance the completeness of the landscape. Therefore, we included our non-coding analyses to the manuscript to provide a better reflection of the value of WGS in general and in this particular study. Changes to the manuscript are already indicated in our response to comment 1a:

Comment 2a

Main findings: authors report as a major achievement the finding that mutations of the FBXW7 gene seems to correlate (in a retrospective analysis) to primary resistance to anti-EGFR therapies. This is actually old news, see Lupini et al., BMC Cancer 2015; 15:808 doi: 10.1186/s12885-015-1752-5 and Korphaisarn K, et al Oncotarget. 2017. doi:10.18632/oncotarget.16848, neither of which are cited in the manuscript.

Response: We apologize for this omission and have now looked into these relevant references. With respect to Lupini et al we greatly add in numbers and significance. More importantly, we have analyzed the actual metastatic lesion and had prospective follow up data whereas Lupini et al analyzed tissue from the primary tumor retrospectively and related this to response of the metastatic disease. We

included this reference in our manuscript. Korphaisarn analyses a mixture of primary and metastatic lesions and shows that *FBXW7* mutations have prognostic value but do not specifically link this to response to (anti-EGFR) treatment. We therefore chose not to include this reference in our manuscript. However, we did add another interesting paper on this topic (PMID 31226844) in which the authors suggest a potential association between *FBXW7* mutations and resistance to anti-EGFR agents. With our data, we were able to confirm this hypothesis. Therefore we added the following text to the manuscript:

Results; Association between molecular landscape and treatment outcome [lines 236-239]:

This suggests that, next to somatic KRAS mutations, somatic *FBXW7* may provide an additional negative selection marker for anti-EGFR treatment. This finding is in concordance with previous reports on *FBXW7* mutation prevalence in non-responding patients on anti-EGFR treatment^{29, 30}.

Discussion [lines 296-300]

In addition, *FBXW7* mutations were predictive for poor response to EGFR-targeted treatments in our prospective cohort. This is in line with previous observations showing that *FBXW7* mutations were enriched in unresponsive patients compared to patients responding well to EGFR-targeted treatments^{29, 30}.

Comment 2b

What one would expect in a Nat Comm paper is (at the very least) the in vitro preclinical validation of FBXW7 mutation(s) as predictor(s) of response to cetuximab/ panitumumab plus, a set of experiments to pinpoint the mechanistic reasons of such a finding.

Response: We agree with the reviewer that this represents an important next step but feel that this is beyond the scope of our current landscape effort.

Comment 3a

One of the most interesting finding in my opinion is that one of the metastases specific signatures (SBS9) is partly associated to the error-prone polymerase. This finding, once again, is reported almost en passant and is not functionally validated by an experiment.

Response: Functional validation of this finding is again beyond our current scope. However, the signature occurs in chronic lymphocytic leukemia and malignant B-cell lymphoma genomes and is characterized by a mutation pattern that has been attributed to POLH (PMID: 29139326). We agree that this is a very interesting finding, especially since POLH activity is associated with anticancer drugs resistance, specifically cisplatin and 5FU, in vitro (PMID: 24259968; PMID: 31064846; PMID: 25125662). Indeed we find that the majority of patients (23 out of 27) with a high SBS9 contribution ($\geq 10\%$) had already received prior treatment. When we look in patients in cluster 3, again the majority of patients (13/15) has received prior treatment. Unfortunately, expression data are available for just 4 samples with high SBS9 contribution, precluding any meaningful analysis with *POLH* expression.

Results - Distinct mutational signature patterns in mCRC patients [lines 179-182]:

Interestingly, Pol η activity has been associated with anticancer drugs resistance, specifically cisplatin and 5FU, in vitro^{22, 23, 24}. Indeed we find that the majority of patients (13 out of 15) in cluster 3 with a

high SBS9 contribution ($\geq 10\%$) had already received prior treatment, although this did not reach statistical significance (Fisher's exact test $p = 0.07$).

Discussion [lines 279-287]

Remarkably, we also observed an mCRC-specific cluster characterized by signatures which are rarely found in primary CRC and are not associated with any treatment (SBS9/39/41). SBS9 is associated with Pol η activity, an error-prone polymerase encoded by the *POLH* gene which mediates translesion synthesis and is induced by replication stress³⁴. Interestingly, high levels of Pol η have been associated with cancer therapy resistance in vitro^{22, 23, 24}. We did observe that the majority of patients with a high relative SBS9 contribution had already received prior treatment, however, unfortunately, sample numbers were too low to directly associate SBS9 contribution with *POLH* expression in our dataset.

Comment 3b

Comparison and validation of TIL score and IHC in 6 MSI samples vs 6 MSS high TMB vs 6 MSS low TMB should be shown in the manuscript.

Response: We have now included Supplementary Figure 2 showing the TIL score in MSI samples ($n=6$), MSS samples with a high TMB (>10 ; $n=15$) and MSS samples with a lower TMB (<10 ; $n=63$). As shown in this figure (also presented here), the average TIL score in MSI samples is significantly higher compared to both MSS samples with a high TMB and with a low TMB (Kruskal-Wallis test ($p=0.037$) followed by Dunn's pairwise comparison (Benjamini-Hochberg corrected $p=0.012$ and $p=0.021$ for MSI compared to MSS with high and low TMB, respectively). This TIL score is based on mRNA expression of selected, TIL-specific, genes as described in PMID: 26553136. Unfortunately, we do not have tissue sections available for our cohort to perform IHC. However, in the original paper [Massink et al. BMC Cancer, 2015, PMID: 26553136] TILs were scored on H&E slides and samples with high and low numbers were subsequently compared to identify a list of TIL-specific mRNAs used for the TIL score. We have added this information to the manuscript:

Commentary figure 2, Suppl. figure 2 in manuscript

Results - Potential clinical implications [lines 248-254]

Interestingly, we did not observe a significantly higher TIL score in the TMB-high samples ($n=21$) compared to the other samples ($n=63$; MWU; $p=0.39$), whereas the average TIL score in MSI samples is significantly higher compared to both MSS samples with a high TMB and with a low TMB (Kruskal-Wallis test ($p=0.037$) followed by Dunn's pairwise comparison (Benjamini-Hochberg corrected $p=0.012$ and $p=0.021$ for MSI compared to MSS with high and low TMB, respectively) (supplementary figure 2).

Summary for reviewer 1:

We have greatly extended our non-coding analyses, and have shown via various means the additional value of WGS. We have not incorporated in vitro validation studies, but have shown additional results/arguments that we feel clearly shows the power and potential value of using genomics in

metastatic cancer patients. We thank the reviewer for the time invested in the thorough review and constructive comments.

Reviewer 2

In this paper, Wilting and colleagues report WGS of 429 pairs of primary CRC tumours and metastases. There is also transcriptomic data for a subset of these samples. They ran a number of analyses, including correlation with prior treatment of the primary CRC. The stated aim of this study is "to provide a comprehensive description of the molecular landscape of metastatic CRC". Overall, this is an interesting paper that has yielded some new insights. I have some suggestions for improvements and requests for clarifications.

Comment 1

The paper's title is rather vague, it can be improved by being more specific about key finding(s).

Response: In light of this reviewer's opinion and the strict word limit we would propose the following alternative title:

"Whole genome sequencing of metastatic CRC reveals prior treatment effects, metastasis-specific features, and potential clinical utility"

Comment 2

Throughout the paper, the terms "pretreatment" or "prior treatment" are used interchangeably. But they aren't defined explicitly, as far as I can see, so it took a while to know for sure that this refers to treatment of the primary CRC. Clearly stating this at the beginning would help.

Response: We apologize for the confusion and have now included a clear definition at the beginning of the results section to clarify this. For clarity purposes we have now refrained from using both terms interchangeably and only use "prior treatment". We changed this terminology throughout the whole manuscript.

Statement added to method section [lines 320-325]:

Based on the provided information regarding all forms of systemic treatment patients received before the study biopsy took place (further referred to as "prior treatment"), we coded the (groups of) active agents using the following abbreviations: PLAT (oxaliplatin), PYR (fluoropyrimidines), TOP (topoisomerase inhibitor; Irinotecan), +targeted (when bevacizumab or panitumumab was added), CHEMCOM (triplet combination therapy).

Comment 3

"Patients with treatment data not matching standard of care...were excluded..." That's rather nebulous. Please elaborate on what this means specifically?

Response: The CPCT-02 trial was open to all patients with advanced cancer starting a new line of treatment, irrespective of the primary cancer origin. We decided to exclude patients with treatment

data not matching standard of care for colorectal cancer (for example: a patient who was treated with imatinib which is not used in CRC treatment suggests a different tumor origin) to avoid erroneous inclusion of patients suffering from another type of cancer.

We have now clarified this in the Methods section [lines 314-318] as follows:

Upon our data request for all CRC patients thus far, we were provided with the data of all patients registered as metastatic CRC patients included between April 2016 and January 2019 (n = 487). Patients whose treatment data did not match standard of care for colorectal cancer were excluded to avoid erroneous inclusion of patients suffering from another type of cancer (n = 28).

Comment 4

With respect to calling variants from WGS, what is the minimum allele frequency to be called? To what extent are subclonal variants being investigated?

Response: The lowest recorded VAF in our cohort when investigating gene-targeting variants is 2.9%. Many of the reported variants are expected to be subclonal, but this was not specifically investigated.

Comment 5

29% of the CRC samples were not classified into a CMS category. Are these somehow intermediate between categories?

Response: Classification of a sample into a certain CMS class is based on correlation of the samples' gene-expression pattern to that of the predefined the CMS groups. When the sample in question shows a comparable correlation to more than one of the predefined CMS groups, the algorithm will not assign a 'winner' and will label the sample as "NA". It indeed means that the expression pattern of the sample shows likeliness to more than 1 CMS category. As justly remarked by reviewer 3, the algorithm we used was constructed using primary CRC tissue and may therefore not perform optimally on metastatic tissue biopsies. Using a different classifier (CMSCaller; PMID 29192179) indeed reduces the number of samples with an "NA" call from 29% to 15%. See further results at reviewer 3, point 1.

Comment 6

I wonder about rearrangement breakpoints, it seems they should be investigated as part of a comprehensive description?

Response: Rearrangements are reported (Figure 1) but we used the term structural variants instead of rearrangements. In short, we observe specifically sized tandem duplications which are not observed in other cancer types, as well as deletions, of which the latter are associated with common fragile sites and treatment outcome

Comment 7

As far as I can see, de novo NMF mutational signature extract was not done on the WGS data. I think it would be worthwhile to do this, to independently verify prior signatures associated with CRC and to check if there might be other novel signatures mixed in with the known prior signatures.

Response: We did actually perform the requested de novo analysis of both single and double base signatures, however all de novo signatures showed cosine similarities of at least 0.7 with already known signatures. For clarity we have now included a short statement that de novo signature extraction did not yield any results beyond the signatures already described in literature. Therefore we added the following statements regarding de novo signature calling:

Results - The molecular landscape of mCRC [lines 114-116]

De novo signature calling using the Non-negative Matrix Factorization algorithm (NMF)¹⁸ did not identify additional signatures besides the known COSMIC signatures in our cohort.

For the reviewer's convenience a short description of the performed analysis and results is included below:

We used the Non-negative Matrix Factorization algorithm (NMF) to estimate the number of signatures present in our data according to the cophenetic coefficient (see commentary figure 2, x-axis shows the 'factorization rank' i.e. the number of signatures, y-axis shows the cophenetic coefficient). To choose the number of signatures, according to Gaujoux and Seoighe (BMC Bioinformatics 2010, PMID 20598126): "The most common approach is to choose the smallest rank for which cophenetic correlation coefficient starts decreasing". We did consider 4 signatures a too small number to capture all mutational patterns in mCRC and thus used 8 major signatures.

We then calculated the cosine similarity between these 8 potential de novo signatures with the existing known COSMIC signatures. For 7 out of 8 signatures the similarity to known signatures was ≥ 0.8 . The 1 remaining potential de novo signature showed a cosine similarity of 0.70 and 0.72 with the known APOBEC signatures SSB2/SBS13, respectively, was only observed in $n=25$ cases (6%) with at least 20% contribution, and was not associated with prior treatment or treatment outcome. These 25 cases are most found in both the primary-like ($n=8$) and mCRC-specific clusters ($n=9$).

Comment 8

"Remarkably, no mutations associated with specific pretreatments or with pretreatment in general were found." This means that mutations in the primary CRC samples were a subset of the mutations observed in the mets, correct? That's not necessarily a surprising result, considering the mets are genetically descended from the primaries, or am I missing something?

Response: What we actually tried to express with this sentence is that compared to untreated metastatic CRC no specific mutations were found enriched in any of the prior treatment groups or in the total group receiving prior treatment. So even though we do clearly observe an increase in the total number of mutations in patients receiving prior treatment compared to untreated mCRC patients this is not reflected in a specific enrichment of particular mutations.

We have now clarified this in the text [lines 131-133]:

Remarkably, even though TMB was increased in patients who received prior treatment compared to treatment-naive patients, no specific mutations (coding or non-coding) were associated with any of the defined prior treatment groups or with prior treatment in general.

Comment 9

The phrase "no non-driver genes with an increased frequency" on line 275 is hard to follow. A reader can figure out what is meant by looking at the table, but it's better to rewrite that long sentence and split it up into two sentences.

We have now rewritten the phrase as follows [lines 152-160]

Mutation frequencies per gene were compared between primary CRC (TCGA cohort) and our total metastatic cohort. For this purpose we selected genes mutated in primary CRC (TCGA cohort) with >5% prevalence and complemented these with here identified metastatic driver genes regardless of their prevalence in primary CRC. Increased frequencies were only observed in driver genes *TP53*, *ZFP36L2*, *KRAS*, and *APC* (Fisher exact test, $FDR \leq 0.012$). A decreased frequency was observed for 21 non-driver genes (Suppl. Table 5) and only 1 driver gene, namely *PIK3CA* (Table 2). With respect to the identified putative non-coding drivers (Table 3), except *PIPSL* and *PTENP1* all of them were enriched in mCRC compared to primary CRC (ICGC dataset; Fisher exact test, $FDR < 5.74E-4$).

Comment 10

To what extent are TILs contributing to the SBS9 observed in cluster 3?

Response: We calculated the Spearman correlation between the TIL score (a proxy for the number of TILs in the sample) and the contribution of SBS9 for the samples in cluster 3 ($R_s = -0.5385$; $p = 0.07$) as well as for all samples in our cohort ($R_s = -0.11$; $p = 0.33$). These results indicate that TILs do not associated to the detected SBS9 mutations and, in addition, that SBS9 mutations do not appear to attract TILs. Due to the limited space available we have not incorporated these results in the manuscript.

Comment 11

For Figure 5, what are the cutoffs for higher and lower % contribution? Why not use a continuous colour spectrum instead of imposing arbitrary cutoffs?

Response: Continuous data were used to generate the heatmap in Figure 5. The contribution percentages were median centered per signature and subsequently values are scaled from red (above median) to yellow (below median). It may be not that clear, but a colour spectrum is present in Figure 5.

We have changed the legend to better reflect this. [figure 5 legend]

Figure 5: Unsupervised hierarchical clustering of metastatic CRC using relative contribution of pre-selected mutational signatures

Heatmap representing the median-centered relative contribution of mutational signatures between samples. Values were scaled from red (relative contribution above median) to yellow (relative contribution below median). Included single and doublet base signatures (SBS/DBS) are indicated at the right. Grouping of mCRC is shown by the dendrogram at the top.

Comment 12

Are there distinguishing features for the set of 28 "MSI genes"? GO enrichment? Enrichment of protein domains, DNA motifs?

Response: We analyzed the gene set for enrichment: after multiple testing correction 'EGF-like domain' (Uniprot-keyword) was enriched ($p=0.001$) compared to human genome as background, found in 6 genes (*TNXB*, *LRP1*, *FAT1*, *STAB2*, *LTBP2* and *LRP2*). The second enriched term was from Gene Ontology: GO:0005041 low-density lipoprotein receptor activity, found in 3 genes (*LRP1*, *STAB2* and *LRP2*. $p=0.01$). Other significantly enriched features were not found (databases used: OMIM, Uniprot, GO-BP, GO-CC, GO-MF, Biocarta and KEGG). We were unable to extract useful knowledge from these associations between MSI and EGF-like domain/lipoprotein receptor activity, especially since 22 out of the 28 genes do not show these characteristics. We chose to not include these results in the text.

Comment 13

It seems the analysis of association between molecular features and treatment results would be a suitable problem for machine learning. The authors should try using some off the shelf ML algorithms and compare them to their current results.

Response: Notwithstanding the great promise of machine learning for decision making processes in medicine, we are convinced our current dataset is not well suited for this type of analysis. As nicely reviewed by Deo in PMID: 26572668, machine learning requires extensive molecular and clinical data on thousands of patients for the training step alone to reduce the risk of overfitting, after which independent test sets are necessary to properly evaluate the model. Our current cohort is quite heterogeneous both with respect to prior treatment and with respect to the treatment given after taking the biopsy. It is unlikely that a single model is able to predict outcome on all these treatments, further reducing the available numbers of patients. Our current aim was to investigate whether any direct associations were present between molecular features and treatment results for which we used the proper statistics. Future larger studies in unbiased cohorts are needed to establish reliable prediction models for treatment outcome.

Comment 14

The sentence at lines 403-405 is really confusing and should be rewritten.

We have rewritten this to [lines 271-276]

In general, the genomic landscape of CRC remains relatively stable in metastatic disease. However, compared to primary CRC, our metastatic CRC cohort showed significant enrichment for mutations in 4 out of 23 coding and 12 out of 15 non-coding (putative) driver genes. From the identified putative drivers, only mutations in *PIK3CA* were significantly decreased in mCRC. Six of our identified coding driver genes are not present in the current CRC-specific MSK-IMPACT panel, namely *ZFP36L2*, *BCL*, *BCL9L*, *ELF3*, *LMTK3*, and *TGIF1*.

In summary for reviewer 2, besides the suggestion to use machine learning, for which we feel our current dataset is not suitable, we have addressed all requests for additional results/clarifications. We thank the reviewer for the thorough review and constructive comments.

Reviewer 3

Congratulations to Mendelaar for putting together promising study results. The manuscript is technically well written, but major re-analysis of the data is recommended (and consequently formatting of results) for publication in Nature Communications. It is hard for me to identify take-home messages. WGS and RNAseq in over 400 clinically annotated metastatic CRC cases and someone would expect more "discoveries". It may be possible, given that genomics is relatively stable over time (and RNAseq was not explored in depth).

Response: We are pleased that the reviewer recognizes the effort put into this manuscript. We agree with the reviewer that our take-home messages need more emphasis and have made a number of changes in the manuscript to achieve this. On the other hand, we feel that we have thoroughly analyzed many different aspects of the current dataset and as such the perceived "lack of discoveries" indeed likely reflects the relative genomic stability over time, which represents a take-home message on its own and was not shown for mCRC to this extent before.

Comment 1

In terms of methodology, CMS classification with CMSclassifier in metastatic tissues is not accurate. You may use CMScaller, which uses mostly cancer cell intrinsic signals (microenvironment of metastases are different)

Response: Indeed the algorithm we used was constructed using primary CRC tissue and may therefore not perform optimally on metastatic tissue biopsies with a different microenvironment. Therefore, as suggested by the reviewer, we now also used CMSCaller (PMID:29192179), which indeed reduces the number of samples with an "NA" call from 29% to 15%. The number of CMS1 and CMS4 calls increased with this algorithm, which in case of CMS1 was not supported by the MSI status of these samples.

We have updated the results section to reflect the new results [lines 65-73].

Indeed, using the alternative CMSCaller algorithm, which is less dependent on signals from the tumor microenvironment, reduced the number of unclassified samples to 14 (15%), whereas still only 3 samples were classified as CMS3¹¹. Twenty-three samples were classified as CMS1, 25 as CMS2, 3 as CMS3, and 27 as CMS4. Regardless of the calling algorithm used, the estimated tumor cell percentage was significantly lower in biopsies classified as CMS4 than in the other subtypes (medians CMS1: 52.5% and 45%; CMS2 61% and 61%; CMS3: none and 66% and CMS4: 34.5% and 42%; KWH; $p=0.0007$ and $p=0.0156$ for CMS Classifier and CMSCaller, respectively), which fits with the described high-stroma content in this subtype³.

Comment 2

Explore additional immune cell infiltration signatures from RNAseq, beyond TILs, and compare with patterns seen in primary CRC.

Response: We have run cibersort (Newman et al, Nature Methods PMID 25822800), a tool to estimate immune cell subsets from expression profiles. The relative contribution of the 22 cell types that were studied do not show any significant association with the cluster groups, MSI vs MSS, biopsy-site or high TMB. A very modest increase of Neutrophils from median 0% in patients that were prior-treated to median 0.7% in patients that were not prior-treated was significant (multiple testing corrected Mann-Whitney $p=0.015$, see commentary figure 3). We deemed the results too unassuming to use in the main text, and since hardly any associations are observed, comparison to primary CRC does not seem very useful.

Commentary figure 3

Comment 3a

The mutational signatures results are difficult to interpret. Any relevant association beyond primary versus metastasis, anti-EGFR exposure?

Response: In our view, we have presented the results regarding mutational signatures as appropriate. Associations between signatures and other parameters, be it mutations, disease state, clinical outcome or other available data, were all statistically evaluated. Perhaps due to the stringent, but necessary, multiple testing corrections only a few of these parameters significantly associated with the signatures. The ones that were significant are presented.

Comment 3b

Does number of treatment lines impact shifts in mutation signatures?

Response: As the provided clinical data was limited, we could only estimate the number of treatment lines based on the dates on which treatment combinations were given. We observed an impact of the number of treatment lines when looking at TMB. The more prior treatment regimens a patient had received, the higher the TMB was. For specific mutational signatures we could not reliably investigate this since we already show that specific treatments induce specific signatures. So therefore we need to stratify patients according to received prior treatments when trying to compare signatures between patients receiving different numbers of treatment lines, resulting in insufficient numbers for reliable comparisons. We did not include this in the manuscript.

Comment 3c

Have you looked at the impact of different signatures on survival in the metastatic setting?

Response: We did analyse the impact of different signatures on treatment response as we incorporated all relevant mutational signatures as a variable into our LASSO analysis. If the reviewer refers to overall survival, we unfortunately do not have OS as clinical parameter.

Comment 4a

The clinical correlates of the study are very poor. Theoretical actionability is not insightful (very few places in the world can execute WGS for clinical decision). I suspect your pragmatic actionability is very low. So my suggestion is to explore the added value of WGS over targeted panels (Foundation Medicine-like)?

Response: We agree with the reviewer on the fact that WGS probably will not function as a common tool used in the clinical decision making process worldwide in short time. We used WGS as a foundation, investigating which candidates are of clinical interest. The currently available therapeutic options that are based on biomarkers would be a viable option for 55% of the patients in our cohort. This clearly indicates additional biomarkers are necessary. Targeted approaches that depend on current knowledge will not give an answer for these patients. As all reviewers pointed out that the added value of WGS was not clear enough, we therefore have emphasized this throughout the manuscript. See also reviewer 1, point 1 for further details on this topic. In short, we acknowledge the fact that targeted panels in the future may serve as a more practical clinical decision making tool. However, we strongly believe in the value of the fundamental knowledge gained through WGS to guide these more targeted research approaches.

Comment 4b

In addition, I suggest you look more carefully at other endpoints - super responders to anti-EGFR therapy versus non-responders? Any other outlier clinical scenario where deep molecular profiling may add value?

Response: These type of analyses would definitely add to our manuscript, but unfortunately the available clinical data was limited. Therefore we could not analyse the different outlier scenarios as suggested.

In summary for reviewer 3: we have elaborated and included additional results upon request of the reviewer, and have put more emphasis on the salient results to clarify the take-home messages. We thank the reviewer for the time invested and the critical evaluation.

REVIEWERS' COMMENTS

Reviewer #1 (Remarks to the Author):

The authors have responded to all my queries satisfactorily and extended their analysis according to suggestions. They also made a quite good case for not incorporating validation studies at this stage of their discovery. I truly thank the authors for the further time invested and for their poised responses.

As far as I am concerned the manuscript is now brilliantly suited for publication.

Reviewer #2 (Remarks to the Author):

The authors have done well enough to clarify and support their claims. No glaring flaws, but more could be done to look at these data in greater depth, e.g., analyzing rearrangement breakpoints at the nucleotide level, investigating clonality, trying machine learning (which doesn't necessarily require a bigger dataset to be applied), as well as points raised by reviewer 1. The authors seem content to stand pat with their story essentially as is, which somewhat undercuts their stated aspiration of doing a comprehensive analysis.

Reviewer #3 (Remarks to the Author):

The authors have done a good job addressing reviewers' comments. My only suggestions is to de-emphasize the clinical utility of the WGS approach (particularly in the abstract). This is a retrospective discovery cohort with many selection biases and authors cannot quantify the added value of WGS over standard-of-care genomic testing. The clinical actionability of 55% (largely RAS wild type...) and FBXW7 mutations as predictors of anti-EGFR (based on a handful of cases when the majority of your RAS/BRAF wild-types have not received targeted therapy) are not robust to appear in the abstract as new insights. Your abstract should be focused on mutational signatures and inferred temporal genomics evolution instead. Please improve description of figures and table legends. Remove the word "outcome" throughout the text as you are not looking at survival data, but responses/benefit to/with standard therapies.

Response to the referees

Reviewer 1

The authors have responded to all my queries satisfactorily and extended their analysis according to suggestions. They also made a quite good case for not incorporating validation studies at this stage of their discovery. I truly thank the authors for the further time invested and for their poised responses. - As far as I am concerned the manuscript is now brilliantly suited for publication.

Authors response: We thank the reviewer for the time invested and the positive feedback.

Reviewer 2

The authors have done well enough to clarify and support their claims. No glaring flaws, but more could be done to look at these data in greater depth, e.g., analyzing rearrangement breakpoints at the nucleotide level, investigating clonality, trying machine learning (which doesn't necessarily require a bigger dataset to be applied), as well as points raised by reviewer 1. The authors seem content to stand pat with their story essentially as is, which somewhat undercuts their stated aspiration of doing a comprehensive analysis.

Authors response:

We are pleased that the reviewer finds we have clarified and further supported the claims we make in the manuscript and feels we have covered any glaring flaws present in the original manuscript. We agree with the reviewer that more can be done with the data. However, we did add several new analyses to the revised version and we have motivated our reasons for not including additional analyses. With respect to our aspiration of doing a comprehensive analysis, we feel we have lived up to it and would like to underline that in this line of research analysis will never be complete, hence we chose to go for comprehensive in its meaning of 'thorough'.

We thank the reviewer for the time invested and his or her appreciation of the value of the data.

Reviewer 3

The authors have done a good job addressing reviewers' comments. My only suggestions is to de-emphasize the clinical utility of the WGS approach (particularly in the abstract). This is a retrospective discovery cohort with many selection biases and authors cannot quantify the added value of WGS over standard-of-care genomic testing. The clinical actionability of 55% (largely RAS wild type...) and FBXW7 mutations as predictors of anti-EGFR (based on a handful of cases when the majority of your RAS/BRAF wild-types have not received targeted therapy) are not robust to appear in the abstract as new insights. Your abstract should be focused on mutational signatures and inferred temporal genomics evolution instead. Please improve description of figures and table legends. Remove the word "outcome" throughout the text as you are not looking at survival data, but responses/benefit to/with standard therapies.

Authors response:

We have de-emphasized the clinical utility in the title and abstract and focussed more on the mutational signatures instead. We do still mention FBXW7 in the abstract as it is a result we observed, and is corroborated by previously published data. We have clarified the description of figures and table legends, and replaced outcome with response or benefit where applicable.

We thank the reviewer for the time invested and the positive feedback.